# Gradient Tree Boosting for Regression Transfer

**Dag Björnberg**  *dag.bjornberg@lnu.se*
*Softwerk AB*
*Department of Computer Science and Media Technology*
*Linnæus University*

**Jonas Nordqvist**  *jonas.nordqvist@lnu.se*
*Department of Mathematics and Physics*
*Linnæus University*

**Morgan Ericsson**  *morgan.ericsson@lnu.se*
*Department of Computer Science and Media Technology*
*Linnæus University*

**Johan Lindeberg**  *johan.lindeberg@lnu.se*
*Department of Forestry and Wood Technology*
*Linnæus University*

**Welf Löwe**  *welf.lowe@lnu.se*
*Softwerk AB*
*Department of Computer Science and Media Technology*
*Linnæus University*

**Johan E.S. Fransson**  *johan.fransson@lnu.se*
*Department of Forestry and Wood Technology*
*Linnæus University*

**Reviewed on OpenReview:** *https://openreview.net/forum?id=b29TPa8NPT*

## Abstract

Many real-world modeling problems are hindered by limited data availability. In such cases, *transfer learning* leverages related source domains to improve predictions in a target domain of interest. We extend the classical gradient tree boosting paradigm to a regression transfer algorithm by modeling the weak learner as a sum of two regression trees. The trees are fitted on source data and target data, respectively, and jointly optimized for the target data. We derive optimal coefficients for the model update under the least-squares, the least-absolute-deviation, and the Huber loss functions. We benchmark our approach against boosting-based regression transfer methods in twelve transfer scenarios. The results indicate that our approach constitutes a competitive alternative within the realm of boosting-based regression transfer. Moreover, we provide a theoretical justification as well as empirical validation that our approach is robust under larger domain shifts.[1]

## 1 Introduction

Transfer learning aims to improve a learner in a *target* domain by transferring information from similar but different *source* domains (Weiss et al., 2016). Typically, target data availability is limited, allowing external information from source datasets to improve the performance of the learner.

---

[1]Code available at: `https://github.com/DageBjorne/transfertreeboost`

Deep learning has become the dominant approach to tackle complex machine learning tasks, with impressive results in areas such as large language modeling, computer vision, and image generation. However, for tabular datasets, gradient boosting methods are often competitive in terms of performance (McElfresh et al., 2023; Grinsztajn et al., 2022). Moreover, they generally require shorter training times and offer greater interpretability than neural networks, making them a viable option for tasks involving tabular datasets.

Unlike neural networks, which enable transfer learning by finetuning learned representations, gradient boosting does not naturally support such adaptation. Consequently, gradient boosting techniques for transfer learning have traditionally been instance-based, operating on the union of the source and target datasets while achieving adaptation by reweighting the importance of source and target data samples. These approaches include TrAdaBoost (Dai et al., 2007), a modification of AdaBoost (Freund and Schapire, 1997) that was introduced for transfer learning in classification tasks. TrAdaBoost was adapted in Pardoe and Stone (2010) to handle regression settings, leading to algorithms such as TrAdaBoost.R2 and two-stage TrAdaBoost.R2. More recent gradient boosting algorithms, such as XGBoost (Chen and Guestrin, 2016), have also been adapted to transfer learning scenarios (Fang et al., 2019; Liu et al., 2020).

Our contribution is a model-based approach for regression transfer using gradient tree boosting. The base learner is modeled as a sum of two regression trees, one fitted on the target dataset and one fitted on the source dataset. We optimize this base learner toward the target dataset with the idea that structural information from the source tree can be used to support transfer learning.

We start by reviewing the classical gradient tree boosting paradigm (for regression tasks) as outlined in Friedman (2001) (Section 2). We then extend the theory to include a sum of two regression trees as the base learner, deriving optimal coefficients for the model update at each iteration for the least-squares, least-absolute-deviation, and Huber loss functions (Section 3). In Section 4, we perform a simulation study for our algorithm under these three loss functions. In Section 5, we benchmark our algorithm against other boosting-based regression transfer approaches. Finally, in Section 6 we discuss the key findings and outline future work.

## 2 Preliminaries

Gradient boosting performs numerical optimization in function space by sequentially fitting weak learners to minimize a differentiable loss function (Friedman, 2001; Mason et al., 1999). For a loss function $L$, we try to find the function $F(\mathbf{x})$ that minimizes the joint expectation given by

$$\mathbb{E}_{y,\mathbf{x}}\left[L(y, F(\mathbf{x}))\right] = \mathbb{E}_{\mathbf{x}}\left[\mathbb{E}_y\left[L(y, F(\mathbf{x})) \mid \mathbf{x}\right]\right] = \mathbb{E}_y\left[L(y, F(\mathbf{x})) \mid \mathbf{x}\right].$$

Following the numerical optimization paradigm, the optimal solution is taken to be

$$F^*(\mathbf{x}) = \sum_{m=0}^{M} f_m(\mathbf{x}),$$

where $f_0$ is an initial guess, $f_m$ is an improvement step, or *boost*, computed according to the optimization method, and $M$ denotes the number of boosts.

For steepest-descent, $f_m(\mathbf{x}) = -\rho_m g_m(\mathbf{x})$ where $-g_m(\mathbf{x})$ is the negative gradient evaluated at $F_{m-1}(\mathbf{x}) = \sum_{i=0}^{m-1} f_i(\mathbf{x})$:

$$
\begin{aligned}
-g_m(\mathbf{x}) &= -\left[\frac{\partial \mathbb{E}_y\left[L(y, F(\mathbf{x})) \mid \mathbf{x}\right]}{\partial F(\mathbf{x})}\right]_{F(\mathbf{x})=F_{m-1}(\mathbf{x})} \\
&= -\mathbb{E}_y\left[\frac{\partial\left[L(y, F(\mathbf{x}))\right]}{\partial F(\mathbf{x})} \mid \mathbf{x}\right]_{F(\mathbf{x})=F_{m-1}(\mathbf{x})}
\end{aligned}
$$

and $\rho_m$ is an optimal step size, found via

$$\rho_m = \arg\min_{\rho} \mathbb{E}_y\left[L(y, F_{m-1}(\mathbf{x}) - \rho g_m(\mathbf{x}))\right].$$

For finite data samples $\{\mathbf{x}_i, y_i\}_1^N$, we may consider updates using the data-based negative gradient given by

$$-g_m(\mathbf{x}_i) = - \left[ \frac{\partial L(y_i, F(\mathbf{x}_i))}{\partial F(\mathbf{x}_i)} \right]_{F(\mathbf{x})=F_{m-1}(\mathbf{x})}.$$

However, the data-based gradient does not generalize to new, unseen data. A solution to this is to fit a parameterized function $h(\mathbf{x}; \mathbf{a}_m)$, often referred to as *weak learner* or *base learner*, to the data-based negative gradient using a least-squares approximation. The function $F_m(\mathbf{x})$ can then be updated every iteration using this "generalized" version of the data-based negative gradient as

$$F_m(\mathbf{x}) = F_{m-1}(\mathbf{x}) + \rho_m h(\mathbf{x}; \mathbf{a}_m).$$

A $J$-terminal node regression tree partitions the input space into finitely many disjoint regions $\{R_j\}_1^J$ and assigns a constant $b_j$ to each region (typically the mean of the responses in that region). When the base learner is a $J$-terminal node regression tree, it can be expressed as

$$h(\mathbf{x}; \mathbf{a}_m) = h(\mathbf{x}; \{b_{jm}, R_{jm}\}_1^J) = \sum_{j=1}^J b_{jm} \mathbf{1}(\mathbf{x} \in R_{jm}).$$

The update is then given by

$$F_m(\mathbf{x}) = F_{m-1}(\mathbf{x}) + \rho_m \sum_{j=1}^J b_{jm} \mathbf{1}(\mathbf{x} \in R_{jm}). \tag{1}$$

In this case, the fit can be improved by considering individual optimal coefficients $\{\gamma_{jm}\}_1^J$ instead of an optimal global step size $\rho_m$, rewriting (1) as

$$F_m(\mathbf{x}) = F_{m-1}(\mathbf{x}) + \sum_{j=1}^J \gamma_{jm} \mathbf{1}(\mathbf{x} \in R_{jm}), \tag{2}$$

where $\gamma_{jm} = \rho_m b_{jm}$. These optimal coefficients are given by

$$\{\gamma_{jm}\}_1^J = \underset{\{\gamma_j\}_1^J}{\arg\min} \sum_{i=1}^N L\left(y_i, F_{m-1}(\mathbf{x}_i) + \sum_{j=1}^J \gamma_j \mathbf{1}(\mathbf{x}_i \in R_{jm})\right).$$

Since we have disjoint regions, for each $\gamma_{jm}$, this reduces to

$$\gamma_{jm} = \underset{\gamma}{\arg\min} \sum_{\mathbf{x}_i \in R_{jm}} L(y_i, F_{m-1}(\mathbf{x}_i) + \gamma).$$

The region-wise optimal coefficients are easily derived in closed form for the least-squares loss and for the least-absolute-deviation loss; for the Huber loss, they can be approximated using standard iterative methods (Friedman, 2001).

To avoid overfitting, the fitted tree is often regularized with a *shrinkage parameter*, or *learning rate $\nu$*. Putting all of this together, we get the algorithm for gradient tree boosting, outlined in Algorithm 1.

## 3 Sum of trees as base learner

We consider our base learner to be the sum of $S \geq 2$ regression trees, where each tree is either constructed from a source dataset, or the target dataset. However, to simplify the exposition in the following section, we have chosen to study the case of $S = 2$ (*i.e.*, a single-source dataset). The generalized version for $S \geq 2$ is covered in Appendix C. The reasoning for the case of $S \geq 2$ mirrors the arguments in this section.

---

**Algorithm 1** TreeBoost

---

1: $F_0(\mathbf{x}) = c_0 = \arg\min_c \sum_{i=1}^N L(y_i, c)$
2: **for** $m = 1, \ldots, M$ **do**
3: $\quad \tilde{y}_i = -\left[\frac{\partial L(y_i, F(\mathbf{x}_i))}{\partial F(\mathbf{x}_i)}\right]_{F(\mathbf{x})=F_{m-1}(\mathbf{x})}, \quad i = 1, \ldots, N$
4: $\quad \{R_{jm}\}_1^J = J\text{-node regression tree} \left(\{\mathbf{x}_i, \tilde{y}_i\}_1^N\right)$
5: $\quad \gamma_{jm} = \arg\min_\gamma \sum_{\mathbf{x}_i \in R_{jm}} L\left(y_i, F_{m-1}(\mathbf{x}_i) + \gamma\right), \quad j = 1, \ldots, J$
6: $\quad F_m(\mathbf{x}) = F_{m-1}(\mathbf{x}) + \nu \sum_{j=1}^J \gamma_{jm} \mathbf{1}(\mathbf{x} \in R_{jm})$
7: **end for**

---

The assumption of our approach is that the source and target domains share informative partition boundaries in the feature space. Mechanically, the tree fitted on the source information acts as a structural prior. Thus, to enable transfer learning, we consider the sum of two regression trees as the base learner

$$h(\mathbf{x}; \{b_{jm}, R_{jm}\}_1^J, \{\hat{b}_{km}, \hat{R}_{km}\}_1^K) = \sum_{j=1}^J b_{jm} \mathbf{1}(\mathbf{x} \in R_{jm}) + \sum_{k=1}^K \hat{b}_{km} \mathbf{1}(\mathbf{x} \in \hat{R}_{km}),$$

where $\{b_{jm}, R_{jm}\}_1^J$ are constructed from the target dataset and $\{\hat{b}_{km}, \hat{R}_{km}\}_1^K$ from the source dataset. Analogous to (1) and (2), the update at iteration $m$ can be improved using individual optimal coefficients. In other words, we consider updates according to

$$F_m(\mathbf{x}) = F_{m-1}(\mathbf{x}) + \sum_{j=1}^J \gamma_{jm} \mathbf{1}(\mathbf{x} \in R_{jm}) + \sum_{k=1}^K \hat{\gamma}_{km} \mathbf{1}(\mathbf{x} \in \hat{R}_{km}),$$

where $\gamma_{jm} = \rho_m b_{jm}$ and $\hat{\gamma}_{km} = \rho_m \hat{b}_{km}$. For each iteration $m$, the individual optimal coefficients are given by

$$\{\{\gamma_{jm}\}_1^J, \{\hat{\gamma}_{km}\}_1^K\} = \underset{\{\{\gamma_j\}_1^J, \{\hat{\gamma}_k\}_1^K\}}{\arg\min} \sum_{i=1}^N L\left(y_i, F_{m-1}(\mathbf{x}_i) + \sum_{j=1}^J \gamma_j \mathbf{1}(\mathbf{x}_i \in R_{jm}) + \sum_{k=1}^K \hat{\gamma}_k \mathbf{1}(\mathbf{x}_i \in \hat{R}_{km})\right). \quad (3)$$

We further consider an additional tunable *weight parameter* $\alpha_m$, determining the contribution from each tree. This yields the following update for $F_m$:

$$F_m(\mathbf{x}) = F_{m-1}(\mathbf{x}) + (1 - \alpha_m) \sum_{j=1}^J \gamma_{jm} \mathbf{1}(\mathbf{x} \in R_{jm}) + \alpha_m \sum_{k=1}^K \hat{\gamma}_{km} \mathbf{1}(\mathbf{x} \in \hat{R}_{km}), \quad (4)$$

where $\alpha_m \in [0, 1]$ may depend on the iteration $m$. This allows both for static and decaying $\alpha_m$, implying that we can control the learning rates for the two terms of the base learner.

Combining all of this, we get our transfer learning algorithm, outlined in Algorithm 2.

In Mason et al. (1999) it was proven that, under mild assumptions, an equivalent formulation of Algorithm 1 will converge. In particular, the assumptions are that the loss function $L$ is convex and has Lipschitz continuous derivative, that the learning rate $\nu$ is sufficiently small, and lastly that the base learner $h(\cdot, \mathbf{a}_m)$ maximizes $-\langle \nabla L(F_{m-1}), h(\cdot, \mathbf{a}_m) \rangle$. The following result indicates that our algorithm possesses the same convergence property as it is a relaxation of the optimization formulation in Algorithm 1.

**Theorem 1.** *TransferTreeBoost (Algorithm 2) admits a convergence rate bound that is superior or equal to that of standard TreeBoost (Algorithm 1).*

A proof of Theorem 1 is given in Appendix A. Moreover, we show in Appendix B that the sum of two regression trees of $J$ and $K$ regions can equivalently be described as a new regression over $J \cdot K$ intersected regions. This shows that the sum of two trees is not merely a heuristic construct, but can be described as a regression tree itself. However, this formulation does not allow for individual weighting of each tree, as formulated in (4).

---

**Algorithm 2** TransferTreeBoost

---

1: Target data $\{\mathbf{x}_i, y_i\}_{i=1}^N$, source data $\{\hat{\mathbf{x}}_i, \hat{y}_i\}_{i=1}^{\hat{N}}$

2: $F_0(\mathbf{x}) = c_0 = \arg\min_c \sum_{i=1}^N L(y_i, c)$

3: **for** $m = 1, \dots, M$ **do**

4: $\quad \tilde{y}_i = -\left[\frac{\partial L(y_i, F(\mathbf{x}_i))}{\partial F(\mathbf{x}_i)}\right]_{F(\mathbf{x})=F_{m-1}(\mathbf{x})}, \quad i = 1, \dots, N$

5: $\quad \tilde{\hat{y}}_i = -\left[\frac{\partial L(\hat{y}_i, F(\hat{\mathbf{x}}_i))}{\partial F(\hat{\mathbf{x}}_i)}\right]_{F(\mathbf{x})=F_{m-1}(\mathbf{x})}, \quad i = 1, \dots, \hat{N}$

6: $\quad \{R_{jm}\}_1^J = J$-node regression tree $\left(\{\mathbf{x}_i, \tilde{y}_i\}_1^N\right)$

7: $\quad \{\hat{R}_{km}\}_1^K = K$-node regression tree $\left(\{\hat{\mathbf{x}}_i, \tilde{\hat{y}}_i\}_1^{\hat{N}}\right)$

8:

$$\{\{\gamma_{jm}\}_1^J, \{\hat{\gamma}_{km}\}_1^K\} = \underset{\{\{\gamma_j\}_1^J, \{\hat{\gamma}_k\}_1^K\}}{\arg\min} \sum_{i=1}^N L\left(y_i, F_{m-1}(\mathbf{x}_i) + \sum_{j=1}^J \gamma_j \mathbf{1}(\mathbf{x}_i \in R_{jm}) + \sum_{k=1}^K \hat{\gamma}_k \mathbf{1}(\mathbf{x}_i \in \hat{R}_{km})\right)$$

9:

$$F_m(\mathbf{x}) = F_{m-1}(\mathbf{x}) + \nu\left((1 - \alpha_m)\sum_{j=1}^J \gamma_{jm}\mathbf{1}(\mathbf{x} \in R_{jm}) + \alpha_m \sum_{k=1}^K \hat{\gamma}_{km}\mathbf{1}(\mathbf{x} \in \hat{R}_{km})\right)$$

10: **end for**

---

*Remark* 1. Since Algorithm 2 jointly optimizes its coefficients with respect to the loss evaluated on the target data in (3), the optimal coefficients $\{\hat{\gamma}_{km}\}_1^K$ of the source tree can be adapted to the degree of domain mismatch. In particular, under substantial domain shift, the optimization can assign smaller magnitudes to the source tree coefficients, thereby reducing its influence. This data-driven adaptivity is crucial for mitigating negative transfer, as it allows the model to downweight misleading source information without requiring additional heuristics.

### 3.1 Deriving optimal coefficients

We will consider three approaches: least-squares (LS) regression, least-absolute-deviation (LAD) regression, and robust regression with the Huber loss. To remain consistent with the terminology used in Friedman (2001), we will refer to these methods as `LSTransferTreeBoost`, `LADTransferTreeBoost`, and `MTransferTreeBoost`, respectively. The region-wise optimal coefficients in (3) cannot be derived in closed form in any of these cases. However, we will provide recipes for how they can be estimated.

### 3.1.1 LS regression

Let $B$ denote our objective, such that $\arg\min B$ corresponds to the RHS of (3), *i.e.*

$$B := \sum_{i=1}^N L\left(y_i, F_{m-1}(\mathbf{x}_i) + \sum_{j=1}^J \gamma_j \mathbf{1}(\mathbf{x}_i \in R_{jm}) + \sum_{k=1}^K \hat{\gamma}_k \mathbf{1}(\mathbf{x}_i \in \hat{R}_{km})\right),$$

which, by symmetry, can be written as the following two double sums

$$B = \sum_{j=1}^J \sum_{\mathbf{x}_i \in R_{jm}} L\left(y_i, F_{m-1}(\mathbf{x}_i) + \gamma_j + \sum_{k=1}^K \hat{\gamma}_k \mathbf{1}(\mathbf{x}_i \in \hat{R}_{km})\right)$$

$$= \sum_{k=1}^K \sum_{\mathbf{x}_i \in \hat{R}_{km}} L\left(y_i, F_{m-1}(\mathbf{x}_i) + \sum_{j=1}^J \gamma_j \mathbf{1}(\mathbf{x}_i \in R_{jm}) + \hat{\gamma}_k\right).$$

By differentiating $B$ w.r.t. $\gamma_j$ and $\hat{\gamma}_k$ for each $j$ and $k$, respectively, we get

$$\frac{\partial B}{\partial \gamma_j} = \sum_{\mathbf{x}_i \in R_{jm}} \frac{\partial L}{\partial \gamma_j}, \quad \text{and} \quad \frac{\partial B}{\partial \hat{\gamma}_k} = \sum_{\mathbf{x}_i \in \hat{R}_{km}} \frac{\partial L}{\partial \hat{\gamma}_k}.$$

If $L$ is the least-squares loss, we need the following conditions to hold for optimal values of $\{\{\gamma_{jm}\}_1^J, \{\hat{\gamma}_{km}\}_1^K\}$:

$$\frac{\partial B}{\partial \gamma_j} = \sum_{\mathbf{x}_i \in R_{jm}} \left( y_i - F_{m-1}(\mathbf{x}_i) - \gamma_j - \sum_{k=1}^{K} \hat{\gamma}_k \mathbf{1}(\mathbf{x}_i \in \hat{R}_{km}) \right) = 0,$$

and

$$\frac{\partial B}{\partial \hat{\gamma}_k} = \sum_{\mathbf{x}_i \in \hat{R}_{km}} \left( y_i - F_{m-1}(\mathbf{x}_i) - \hat{\gamma}_k - \sum_{j=1}^{J} \gamma_j \mathbf{1}(\mathbf{x}_i \in R_{jm}) \right) = 0.$$

Let $|R_{jm}|$ and $|\hat{R}_{km}|$ denote the amount of datapoints contained in regions $R_{jm}$ and $\hat{R}_{km}$, respectively, and let $N_{R_{jm} \cap \hat{R}_{km}} = N_{\hat{R}_{km} \cap R_{jm}}$ denote the amount of datapoints in the intersection of $R_{jm}$ and $\hat{R}_{km}$. Furthermore, put

$$\mathbf{R} := \begin{bmatrix} |R_{1m}| & 0 & \cdots & 0 \\ 0 & |R_{2m}| & \cdots & 0 \\ \vdots & \vdots & \ddots & \vdots \\ 0 & 0 & \cdots & |R_{Jm}| \end{bmatrix}, \quad \widehat{\mathbf{R}} := \begin{bmatrix} |\hat{R}_{1m}| & 0 & \cdots & 0 \\ 0 & |\hat{R}_{2m}| & \cdots & 0 \\ \vdots & \vdots & \ddots & \vdots \\ 0 & 0 & \cdots & |\hat{R}_{Km}| \end{bmatrix},$$

$$\mathbf{N} := \begin{bmatrix} N_{R_{1m} \cap \hat{R}_{1m}} & N_{R_{1m} \cap \hat{R}_{2m}} & \cdots & N_{R_{1m} \cap \hat{R}_{Km}} \\ N_{R_{2m} \cap \hat{R}_{1m}} & N_{R_{2m} \cap \hat{R}_{2m}} & \cdots & N_{R_{2m} \cap \hat{R}_{Km}} \\ \vdots & \vdots & \ddots & \vdots \\ N_{R_{Jm} \cap \hat{R}_{1m}} & N_{R_{Jm} \cap \hat{R}_{2m}} & \cdots & N_{R_{Jm} \cap \hat{R}_{Km}} \end{bmatrix}, \quad \text{and} \quad \mathbf{r} := \begin{bmatrix} \sum_{\mathbf{x}_i \in R_{1m}} y_i - F_{m-1}(\mathbf{x}) \\ \sum_{\mathbf{x}_i \in R_{2m}} y_i - F_{m-1}(\mathbf{x}) \\ \vdots \\ \sum_{\mathbf{x}_i \in R_{Jm}} y_i - F_{m-1}(\mathbf{x}) \\ \sum_{\mathbf{x}_i \in \hat{R}_{1m}} y_i - F_{m-1}(\mathbf{x}) \\ \sum_{\mathbf{x}_i \in \hat{R}_{2m}} y_i - F_{m-1}(\mathbf{x}) \\ \vdots \\ \sum_{\mathbf{x}_i \in \hat{R}_{Km}} y_i - F_{m-1}(\mathbf{x}) \end{bmatrix}.$$

Now, combine $\mathbf{N}$, $\mathbf{R}$, and $\widehat{\mathbf{R}}$ into a block matrix

$$\mathbf{M} := \begin{bmatrix} \mathbf{R} & \mathbf{N} \\ \mathbf{N}^\mathsf{T} & \widehat{\mathbf{R}} \end{bmatrix}. \tag{5}$$

The optimal coefficients can thus be found by solving the system of equations given by

$$\mathbf{M} \begin{bmatrix} \gamma \\ \hat{\gamma} \end{bmatrix} = \mathbf{r}. \tag{6}$$

The following proposition shows that $\mathbf{M}$ is singular. Consequently, we will rely on least-squares approximations to obtain estimates. The singularity of $\mathbf{M}$ arises because the union of regions in both the target and the source trees covers the same ambient space, which creates a linear dependence in the block matrix. An alternative solution, which may yield unique solutions, is to introduce a small ridge regularization.

**Proposition 1.** *Let $\mathbf{M}$ be defined as in (5). Then $\mathbf{M}$ is singular.*

A proof is given in Appendix A.

### 3.1.2 LAD regression

Let the loss $L$ in (3) be the least-absolute-deviation loss. Then the optimal coefficients at iteration $m$ are given by

$$\{\{\gamma_{jm}\}_1^J, \{\hat{\gamma}_{km}\}_1^K\} = \underset{\{\{\gamma_j\}_1^J, \{\hat{\gamma}_k\}_1^K\}}{\arg\min} \sum_{i=1}^{N} \left| y_i - F_{m-1}(\mathbf{x}_i) - \sum_{j=1}^{J} \gamma_j \mathbf{1}(\mathbf{x}_i \in R_{jm}) - \sum_{k=1}^{K} \hat{\gamma}_k \mathbf{1}(\mathbf{x}_i \in \hat{R}_{km}) \right|.$$

Now, introduce auxiliary slack variables defined by

$$t_i := \left| y_i - F_{m-1}(\mathbf{x}_i) - \sum_{j=1}^{J} \gamma_j \mathbf{1}(\mathbf{x}_i \in R_{jm}) - \sum_{k=1}^{K} \hat{\gamma}_k \mathbf{1}(\mathbf{x}_i \in \hat{R}_{km}) \right|, \quad i = 1, \dots, N. \tag{7}$$

Then we can formulate the search for the optimal coefficients as a linear programming (LP) problem as follows:

$$\min \sum t_i \tag{8}$$

$$\text{s.t.} \quad t_i \geq \left| y_i - F_{m-1}(\mathbf{x}_i) - \sum_{j=1}^{J} \gamma_j \mathbf{1}(\mathbf{x}_i \in R_{jm}) - \sum_{k=1}^{K} \hat{\gamma}_k \mathbf{1}(\mathbf{x}_i \in \hat{R}_{km}) \right|, \quad i = 1, \dots, N.$$

This is an optimization problem with $N + J + K$ unknown variables and $2N$ linear constraints. Let $\mathbf{I}_N$ be the identity matrix of size $N$, $\mathbf{c}_N$ be the identity vector of length $N$, $\mathbf{t}^\intercal = \begin{bmatrix} t_1 t_2 \dots t_N \end{bmatrix}$ be the transpose of the vector to the $N$ slack variables, and $(\mathbf{t}\gamma\hat{\gamma})^\intercal = \begin{bmatrix} \mathbf{t}^\intercal \gamma_1 \gamma_2 \dots \gamma_J \hat{\gamma}_1 \hat{\gamma}_2 \dots \hat{\gamma}_K \end{bmatrix}$ be the transpose of the vector consisting of all decision variables (of length $N + J + K$). Furthermore, put

$$\mathbf{\Gamma} := \begin{bmatrix} \mathbf{1}(\mathbf{x}_1 \in R_{1m}) & \cdots & \mathbf{1}(\mathbf{x}_1 \in R_{Jm}) & \mathbf{1}(\mathbf{x}_1 \in \hat{R}_{1m}) & \cdots & \mathbf{1}(\mathbf{x}_1 \in \hat{R}_{Km}) \\ \mathbf{1}(\mathbf{x}_2 \in R_{1m}) & \cdots & \mathbf{1}(\mathbf{x}_2 \in R_{Jm}) & \mathbf{1}(\mathbf{x}_2 \in \hat{R}_{1m}) & \cdots & \mathbf{1}(\mathbf{x}_2 \in \hat{R}_{Km}) \\ \vdots & \ddots & \vdots & \vdots & \ddots & \vdots \\ \mathbf{1}(\mathbf{x}_N \in R_{1m}) & \cdots & \mathbf{1}(\mathbf{x}_N \in R_{Jm}) & \mathbf{1}(\mathbf{x}_N \in \hat{R}_{1m}) & \cdots & \mathbf{1}(\mathbf{x}_N \in \hat{R}_{Km}) \end{bmatrix},$$

and

$$\mathbf{b} := \begin{bmatrix} y_1 - F_{m-1}(\mathbf{x}_1) \\ y_2 - F_{m-1}(\mathbf{x}_2) \\ \vdots \\ y_N - F_{m-1}(\mathbf{x}_N) \end{bmatrix}.$$

We can write (8) in matrix form as

$$\min \quad \mathbf{c}^\intercal \mathbf{t}$$

$$\text{s.t.} \begin{bmatrix} -\mathbf{I}_N & \mathbf{\Gamma} \\ -\mathbf{I}_N & -\mathbf{\Gamma} \end{bmatrix} (\mathbf{t}\gamma\hat{\gamma}) \leq \begin{bmatrix} \mathbf{b} \\ -\mathbf{b} \end{bmatrix}. \tag{9}$$

Thus, we conclude that the optimal coefficients are found by solving the linear optimization in (9).

### 3.1.3 Robust regression with the Huber loss

The Huber loss function, introduced by Huber (1964), combines the advantages of the least-squares loss and the least-absolute-deviation loss. It is efficient when errors are normally distributed, yet remains robust to outliers. Formally, it is defined by

$$L(y, F) = \begin{cases} \frac{1}{2}(y - F)^2 & \text{if } |y - F| \leq \delta, \\ \delta \left( |y - F| - \frac{\delta}{2} \right) & \text{if } |y - F| > \delta \end{cases}.$$

The threshold $\delta$ serves as a cutoff between inliers and outliers, and is often selected based on a quantile of the residual distribution. In gradient boosting frameworks, it can be re-estimated at each iteration from the current residuals. For the Huber loss function, the optimal coefficients are given by

$$\{\{\gamma_{jm}\}_1^J, \{\hat{\gamma}_{km}\}_1^K\} =$$

$$\underset{\{\{\gamma_j\}_1^J, \{\hat{\gamma}_k\}_1^K\}}{\arg\min} \left[ \sum_{\mathbf{x}_i : |y_i - F_{m-1}(\mathbf{x}_i)| \leq \delta_m} \frac{1}{2} \left( y_i - F_{m-1}(\mathbf{x}_i) - \sum_{j=1}^J \gamma_j \mathbf{1}(\mathbf{x}_i \in R_{jm}) - \sum_{k=1}^K \hat{\gamma}_k \mathbf{1}(\mathbf{x}_i \in \hat{R}_{km}) \right)^2 \right.$$

$$\left. + \sum_{\mathbf{x}_i : |y_i - F_{m-1}(\mathbf{x}_i)| > \delta_m} \delta_m \left( |y_i - F_{m-1}(\mathbf{x}_i) - \sum_{j=1}^J \gamma_j \mathbf{1}(\mathbf{x}_i \in R_{jm}) - \sum_{k=1}^K \hat{\gamma}_k \mathbf{1}(\mathbf{x}_i \in \hat{R}_{km})| - \frac{\delta_m}{2} \right) \right].$$

Now, introduce $N$ slack variables defined as in (7). Then the search for the optimal coefficients $\{\{\gamma_{jm}\}_1^J, \{\hat{\gamma}_{km}\}_1^K\}$ can be formulated as a quadratic programming (QP) problem according to

$$\min \sum_{t_i : |y_i - F_{m-1}(\mathbf{x}_i)| \leq \delta_m} \frac{1}{2} t_i^2 + \sum_{t_i : |y_i - F_{m-1}(\mathbf{x}_i)| > \delta_m} \delta_m t_i$$

$$\text{s.t.} \quad t_i \geq \left| y_i - F_{m-1}(\mathbf{x}_i) - \sum_{j=1}^J \gamma_j \mathbf{1}(\mathbf{x}_i \in R_{jm}) - \sum_{k=1}^K \hat{\gamma}_k \mathbf{1}(\mathbf{x}_i \in \hat{R}_{km}) \right|, \quad i = 1, \ldots, N.$$

Similarly to LAD regression, we obtain an optimization problem with $N + J + K$ unknowns and $2N$ linear constraints. The constraints for Huber become the same as for LAD, but the objective is split into a quadratic part and a linear part.

Let $\mathbf{Q}_N$ be the identity matrix of size $N$, but put the diagonal element to zero if the corresponding $i$th slack variable does not contribute to the quadratic part. Similarly, let $\mathbf{c}_N$ be the identity vector of length $N$, but put the $i$th element to zero if the corresponding slack variable is not in the linear part of the objective. Then we get the QP problem formulated in matrix form as

$$\min \quad \frac{1}{2} \mathbf{t}^\mathsf{T} \mathbf{Q}_N \mathbf{t} + \mathbf{c}_N^\mathsf{T} \mathbf{t}$$

$$\text{s.t.} \begin{bmatrix} -\mathbf{I}_N & \mathbf{\Gamma} \\ -\mathbf{I}_N & -\mathbf{\Gamma} \end{bmatrix} (\mathbf{t}\gamma\hat{\gamma}) \leq \begin{bmatrix} \mathbf{b} \\ -\mathbf{b} \end{bmatrix}. \tag{10}$$

*Remark* 2. Note that the matrix $\mathbf{Q}_N$ is positive semi-definite, which together with linear constraints guarantees convexity of the problem.

### 3.1.4   Note on time complexity

Let $N$ and $\hat{N}$ denote the number of target and source datapoints, respectively, and let $p$ be the number of features. Constructing a single boosting tree as described in Algorithm 1 incurs a computational complexity of $\mathcal{O}(Npd)$, where $d$ denotes the depth of the tree (see, *e.g.*, (Chen and Guestrin, 2016) for a complexity analysis).[2] Henceforth, we will approximate $d$ using the number of terminal regions, assuming $d \approx \log_2(J)$, where $J$ is the number of regions.

For Algorithm 2, the additional cost for each boosting iteration $m$, compared to Algorithm 1, is given by

$$\mathcal{O}\left( \hat{N} p \log_2(K) + \mathcal{C}_{\text{alg}}(N, J, K) \right).$$

Here, the function $\mathcal{C}$ captures the overhead associated with solving the various optimization tasks outlined in previous sections. Specifically, solving the least-squares problem in Section 3.1.1 adds a worst-case cost of $\mathcal{C}_{\text{LS}}(N, J, K) \in \mathcal{O}((J + K)^3)$, since the normal matrix $\mathbf{M}$ is of dimension $J + K$.

---

[2]Mapping to their notation implies $K = 1$, $d = \log_2(J)$, and $||\mathbf{x}||_0 = Np$; assuming pre-sorted features yields our stated complexity.

Let $I$ denote the number of iterations performed for the LP problem described in Section 3.1.2. A naïve implementation of solving the LP problem is, assuming that $N \gg J, K$, upper bounded by $\mathcal{O}(IN^3)$. However, by exploiting the block-diagonal structure of the KKT system, we obtain a worst-case additional cost of solving the LP problem bounded by $\mathcal{C}_{\text{LAD}}(N, J, K) \in \mathcal{O}(I(N(J+K)^2 + (J+K)^3))$ (Boyd and Vandenberghe, 2004, Sec. 11.8.2). Finally, the Huber regression in Section 3.1.3 is formulated as a QP problem with identical dimensions, so the same bound applies for the naïve approach. Applying the same logic and exploiting the block-diagonal structure, solving the QP problem yields a matching worst-case complexity of $\mathcal{C}_{\text{M}}(N, J, K) \in \mathcal{O}(I(N(J+K)^2 + (J+K)^3))$.

We provide the derivations for the latter bounds in Appendix D. In Appendix E we provide wall-clock runtimes for our current implementations.

## 3.2 Implementation

The base learner was implemented using `DecisionTreeRegressor()` (on both the source and target data) in `scikit-learn` (Pedregosa et al., 2011). Implementing the coefficient updates for each method is straightforward given their matrix representations. For LSTransferTreeBoost (system of equations defined in (6)), we use Numpy's built-in least-squares solver. For LADTransferTreeBoost (LP problem defined in (9)), and for MTransferTreeBoost (QP problem defined in (10)), we use the default solver configurations in `Scipy` (Virtanen et al., 2020) and `CVXPY` (Diamond and Boyd, 2016), respectively.

# 4 Simulation study

Assessing our three regression approaches with respect to dataset size, error distribution, and degree of domain shift is most conveniently achieved using simulated datasets, where these properties can be explicitly controlled.

## 4.1 Dataset

We use the *Friedman #1* dataset, introduced in Friedman (1991). The dataset is generated using 10 predictor variables $x_1, \ldots, x_{10}$ drawn from a uniform distribution over the interval $[0, 1]$, with a response generated from the first 5 predictor variables according to

$$y = 10\sin(\pi x_1 x_2) + 20(x_3 - 0.5)^2 + 10x_4 + 5x_5 + \epsilon, \tag{11}$$

where $\epsilon$ is generated according to some error distribution.

For transfer learning purposes, we also construct an *altered* version of the Friedman #1 dataset, generated according to the prescription in Pardoe and Stone (2010). In particular, two types of transformations are introduced to the features, namely *feature scaling* and *feature shift*, obtaining transformed features given by

$$\hat{x}_i = b_i x_i + c_i, \quad i = 1, \ldots, 5.$$

Furthermore, the response variable $y$ is modified with additional scaling and an altered response is obtained according to

$$\hat{y} = a_1 10\sin(\pi \hat{x}_1 \hat{x}_2) + a_2 20(\hat{x}_3 - 0.5)^2 + a_3 10\hat{x}_4 + a_4 5\hat{x}_5 + \hat{\epsilon}, \tag{12}$$

where $\hat{\epsilon}$ is the error distribution for the altered version of the response. The fixed parameters $a_i, b_i$ were drawn from a normal distribution $\mathcal{N}(1, 0.1d)$, whereas $c_i$ were drawn from $\mathcal{N}(0, 0.05d)$, where the positive integer $d$ can be thought of as a "disturbance level", controlling the strength of the domain difference between the source and target datasets.

The target dataset $\{\mathbf{x}_i, y_i\}_{i=1}^N$ and the source dataset $\{\hat{\mathbf{x}}_i, \hat{y}_i\}_{i=1}^{\hat{N}}$ can be generated from (11) and (12), respectively. This way, we can assess performance across train sizes, error distributions, and the closeness of the joint distributions (this "closeness" being determined by the disturbance level $d$).

## 4.2 Experimental setup

We considered three types of distributions of the errors $\epsilon$ and $\hat{\epsilon}$: the normal distribution (well-behaved tails), the slash distribution (very heavy tails), and the $t(2)$-distribution (moderately heavy tails). The errors were adjusted to give a 3/1 signal-to-noise ratio (SNR). We added errors to the target training dataset and the source dataset, but not to the validation or test sets.

The source dataset size was set to a fixed amount of 1000 datapoints, whereas the target dataset consisted of 100, 300, or 500 datapoints. An additional validation set of 1000 datapoints was generated to be used for early stopping. Finally, each hyperparameter configuration was evaluated on an unbiased test set consisting of another 1000 datapoints. This procedure was repeated for disturbance levels $d \in \{3, 6, 9\}$, across 5 distinct random seeds. The model fitting process was interrupted after 5 consecutive boosting rounds (epochs) of a non-improving RMSE on the validation set.

We considered the following hyperparameters:

- Shrinkage parameter $\nu$: We let this be fixed as $\nu = 0.1$.

- Maximum source/target tree depth: We considered maximum depths of source/target trees between 1 and 3. All combinations of these were considered (*e.g.*, source tree depth = 1, target tree depth = 3 is a possible combination).

- Weight parameter $\alpha_m$: We let this be defined by the exponential decay scheduler $\alpha_m = m_0 e^{-km}$, with hyperparameters $m_0 \in \{0.1, 0.5, 0.9\}$, and $k \in \{0, 0.01, 0.05\}$. Note that setting $k = 0$ results in static $\alpha_m$-values equal to $m_0$, whereas $k > 0$ gives a decaying schedule.

- Threshold $\delta_m$ (only for MTransferTreeBoost): Although this parameter might be optimized through experimentation, we set it to be the 90% quantile of the current residual distribution.

## 4.3 Results

We present the RMSE values for the five best hyperparameter configurations for each triplet (target instances, error distribution, $d$) and for each regression technique. These configurations were selected by averaging performance across the five seeds and retaining those with the lowest test RMSE. These results are shown in Figure 1.

Performance-wise, MTransferTreeBoost is the clear winner: it achieves superior results compared to LSTransferTreeBoost and LADTransferTreeBoost for both slash and $t(2)$ errors, and performs on par with LSTransferTreeBoost under normally distributed errors.

Due to its robust performance across settings, we report the top-5 hyperparameter configurations for MTransferTreeBoost for each triplet (target instances, error distribution, $d$). These are shown in Appendix H.1. We did not observe any functional patterns for maximum source/target tree depth or the weight parameter $\alpha_m$ based on the number of target instances or disturbance level $d$. However, we noted that using the static weight parameter $\alpha_m = 0.9$ (*i.e.*, $m_0 = 0.9, k = 0$) consistently resulted in poor performance. Both static values ($\alpha_m = \alpha \leq 0.5$) and decaying schedules occurred in the top-5 configurations.

## 4.4 Illustrative example of transfer effect

As discussed in Remark 1, a core advantage of our approach is that the source tree contribution can be adapted relative to the degree of domain shift, which is a direct consequence of optimizing the loss on the target data. We illustrate this behavior on the Friedman #1 dataset.

For each regression method, we simulated 300 target instances and 1000 source instances with disturbance levels $d \in \{5, 20, 60\}$. To isolate the transfer effect, no noise was added to the data. For all approaches, we used a maximum tree depth of 2 for both source and target trees, a constant weight parameter $\alpha_m = 0.5$, and a shrinkage parameter $\nu = 0.1$. For MTransferTreeBoost, the threshold $\delta_m$ was set to the 90% quantile of the current residual distribution.

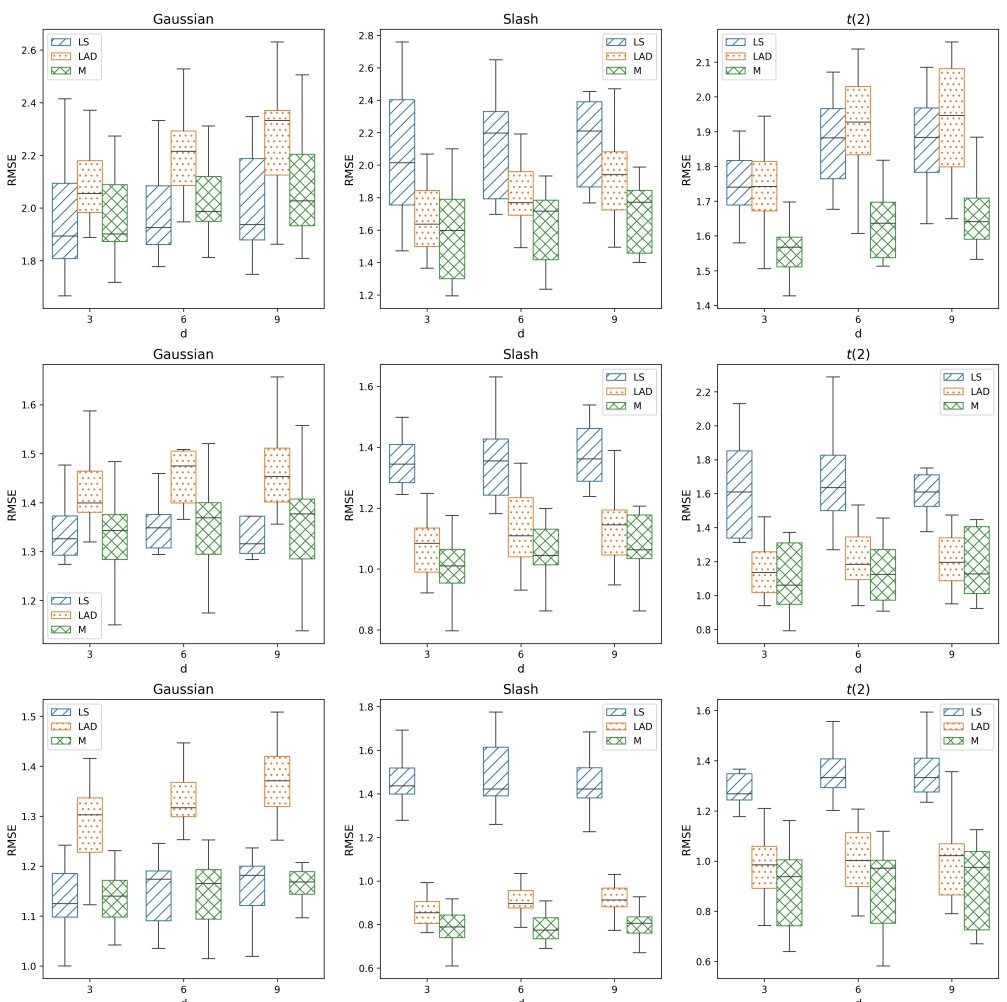

Figure 1: Test RMSE values on the Friedman #1 dataset using the best five hyperparameter settings for each triplet (target instances, error distribution, $d$), for LSTransferTreeBoost (LS), LADTransferTreeBoost (LAD), and MTransferTreeBoost (M). The top, middle, and bottom rows correspond to 100, 300, and 500 target instances, respectively.

To quantify the relative contribution of the source tree, we define

$$q_m := \frac{\sum_k |\hat{\gamma}_{km}|}{\sum_j |\gamma_{jm}|},$$

which measures the total magnitude of the source tree's optimal coefficients relative to those of the target tree at each iteration $m$.

We performed 300 boosting iterations. The values of $q_m$ were averaged over 100 independent experiments and are shown in Figure 2. As the disturbance level $d$ increases, $q_m$ decreases, indicating that the model assigns less weight to the source tree under larger domain shifts. This behavior is consistent with the adaptive mechanism described in Remark 1.

## 5 Benchmarking

In this section, we describe our datasets and benchmark algorithms and present the empirical results. We evaluate our approach against several boosting-based transfer learning algorithms on eight datasets and in

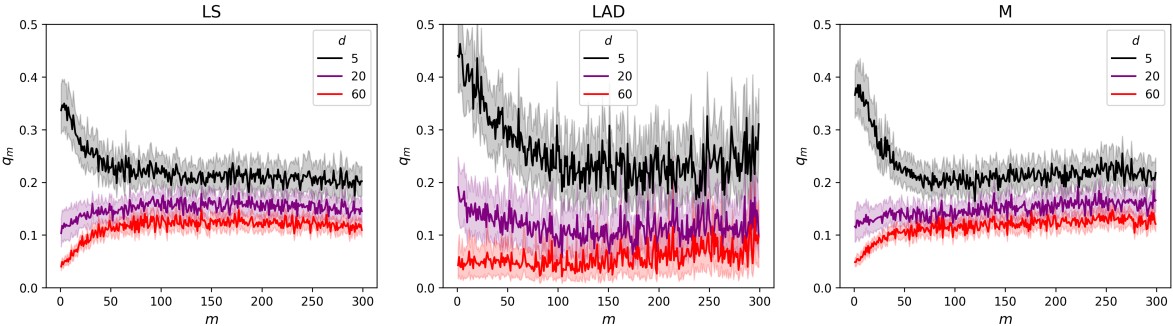

Figure 2: Relative magnitude of source tree contributions, measured by $q_m$, across boosting iterations for different disturbance levels $d$ under LSTransferTreeBoost (LS), LADTransferTreeBoost (LAD), and MTransferTreeBoost (M).

twelve transfer scenarios. Moreover, we demonstrate the robustness of our approach on the Friedman #1 dataset.

## 5.1 Datasets

Monte Carlo simulation studies are convenient for assessing model performance under controlled conditions. However, transfer learning algorithms must be investigated in real-world transfer scenarios. Therefore, we evaluate our approaches on datasets with artificial source–target splits and on two real-world transfer applications in forestry.

### 5.1.1 UCI datasets

We select six regression datasets from the UCI machine learning repository[3]. Because these datasets are from only one domain, the source and target data have to be constructed artificially using splitting techniques. Although such strategies do not necessarily result in meaningful source-target splits, they are commonly used for evaluating transfer methods (Dai et al., 2007; Pardoe and Stone, 2010; Segev et al., 2016).

Following Pardoe and Stone (2010), we create source–target splits based on a continuous feature with a moderate degree of correlation (approximately 0.4) with the response variable. For each dataset, we identify the continuous feature whose absolute correlation with the response is closest to 0.4. Instances are then ordered according to this feature and divided into four equally sized subsets. The first or the last subset is randomly chosen to constitute the target data, and the remaining three quarters are used as source data. Finally, the splitting feature is removed from both source and target datasets. The resulting datasets are summarized in Table 1.

Table 1: UCI datasets after source-target splitting.

| Dataset | Target $N$ | Source $\hat{N}$ | Features |
|---|---|---|---|
| InfraRed | 254 | 764 | 32 |
| Concrete strength | 257 | 773 | 7 |
| Auto MPG | 98 | 294 | 6 |
| Real estate valuation | 103 | 311 | 5 |
| Airfoil self-noise | 378 | 1125 | 4 |
| Forest fires | 130 | 387 | 7 |

---

[3]https://archive-beta.ics.uci.edu/

### 5.1.2 Stem volume

Estimating the stem volume of individual trees is essential for supporting accurate and detailed forest management. Advances in sensing technologies, such as digital imagery and terrestrial laser scanning, make it possible to extract geometric information from the entire stem of individual trees. However, obtaining complete stem measurements typically requires terrestrial data collection, often involving operators walking through the forest and capturing detailed recordings, which is a time-consuming process. Thus, estimating the total stem volume using only the lower part of the stem is an interesting task.

The Swedish stem bank (Björklund and Moberg, 1999; Moberg, 2001) contains measurements of individual trees. Data were collected from 198 Scots pine trees and 114 Norway spruce trees sampled across Sweden. The trees were cut into logs and scanned using imaging technology, providing detailed information about the stems. For example, stem diameters are available at 10 cm intervals, enabling accurate estimation of stem volume.

We use data from the Swedish stem bank to estimate total stem volume based on measurements from the lower part of the stem. A total of 23 predictor variables, corresponding to stem diameters measured between 0.3 m and 2.5 m, were used to predict the entire stem volume.

We considered three distinct transfer scenarios:

1. **Pine → Spruce:** Pine trees constituted the source dataset, and spruce trees the target dataset. This scenario tests transfer across species, which differ in growth patterns and stem shapes. It simulates a common problem in forestry where a well-studied species dataset must inform predictions for another species with limited data.

2. **N. Sweden → S. Sweden**: Trees were split based on geographic location, with the southernmost 25% used as target data and the rest as source data. This scenario captures environmental and regional differences, such as variations in soil, climate, and growth conditions, reflecting the challenge of transferring models across locations.

3. **Pine → Spruce with label shift:** Same as (1), but the source dataset used tree height as the label, while the target dataset used stem volume. This scenario tests transfer across both species and related tasks, where source and target labels are different but correlated.

### 5.1.3 Remote sensing

We consider a remote sensing application in forest variable estimation. National airborne laser scanning (ALS) and national field inventory (NFI) data are commonly used to model forest variables at the national level. A common approach is to summarize ALS point-cloud data into statistical metrics and relate these metrics to field-measured variables, which are typically aggregated as means or totals over NFI plots.

We consider a Swedish source dataset and a Latvian target dataset. A total of 30 ALS-derived metrics were used as predictor variables, with mean stem diameter as the response variable. The Swedish dataset contains 8 412 observations, while the Latvian dataset contains 1 900 observations. Previous ALS-based forest inventory studies have suggested that around 500 field plots can be sufficient to build reliable models (Maltamo et al., 2009).

To evaluate our transfer learning approach, we randomly selected 300 target instances from the Latvian dataset prior to model training. We then considered three distinct transfer scenarios:

1. **Sweden → Latvia:** We randomly selected 2000 datapoints from Sweden to be used as source data.

2. **N. Sweden → Latvia:** We divided Sweden into 4 equally sized regions (dataset wise) from north to south, with only the north as source data. As Latvian forest conditions are more similar to the conditions in southern Sweden, *e.g.* in terms of tree size variability and tree species composition, this scenario represents a larger domain shift compared to (1). We selected 2000 datapoints as source data.

3. **Sweden → Latvia with label shift:** Same as (1), but the source dataset used stem volume as the label, while the target dataset used stem diameter.

## 5.2 Benchmark algorithms

Since our approach builds on gradient boosting, we benchmark it against well-established boosting-based regression transfer methods, which constitute the most relevant comparison class. For reference, we additionally include tabular ResNet architectures (Gorishniy et al., 2021) to contextualize performance relative to deep-learning approaches (see Appendix F). The remainder of this section focuses on the boosting-based benchmark algorithms.

### 5.2.1 XGBoost

A natural baseline for transfer learning is to ignore the source data and train only on the target data. For tabular regression problems, gradient tree boosting methods, such as XGBoost (Chen and Guestrin, 2016), consistently achieve state-of-the-art performance across a wide range of datasets, as demonstrated in McElfresh et al. (2023); Grinsztajn et al. (2022). Consequently, we include XGBoost as a strong non-transfer baseline.

### 5.2.2 XGBoost Warmstart and Pooled XGBoost

We consider two naïve transfer approaches based on XGBoost:

- `XGBoost Warmstart`—In this approach, we first train XGBoost on source data, and then continue boosting on target data.

- `Pooled XGBoost`—Here, we simply combine source and target data into one large dataset used during training. Further, we augment the feature set with a binary indicator, explicitly marking each data point's origin as either source or target.

These approaches serve as simple and practical baselines for transfer learning using gradient tree boosting. Despite their simplicity, such strategies provide relevant reference points as they are straightforward to implement within existing XGBoost pipelines and require no modification of the underlying learning algorithm. Thus, they are commonly employed in multi-source and transfer learning settings, see *e.g.* (Komodromos et al., 2026; Wang et al., 2025; Kebede et al., 2026).

### 5.2.3 TrAdaBoost.R2

TrAdaBoost.R2, described in Pardoe and Stone (2010), extends AdaBoost (Freund and Schapire, 1997) and TrAdaBoost (Dai et al., 2007) to a regression transfer learning setting. It operates as an instance-based approach by iteratively reweighting both source and target samples using an adjusted error scheme. Source instances that are well-predicted tend to gain higher weights, while poorly predicted source instances are progressively downweighted across iterations. In contrast, target instances follow the reverse pattern, where poorly predicted instances are upweighted and well-predicted instances are downweighted.

We implemented TrAdaBoost.R2 using the `Adapt` package (de Mathelin et al., 2021) in Python. In the original study, the M5Prime (Wang and Witten, 1997) regression tree was employed as the base learner. Unlike standard regression trees, M5Prime fits linear models at the leaf nodes rather than predicting constant values. We implemented the M5Prime base learner using the `m5py` package (Marié, 2022).

## 5.3 Experiments

We found that LSTransferTreeBoost performed on par with or better than MTransferTreeBoost in our experiments on real data. Consequently, we only present the results of LSTransferTreeBoost in this section.

### 5.3.1 Hyperparameter tuning

Our approach introduces an additional source tree and an individual tree weighting $\alpha_m$ at each iteration (controlled by $m_0$ and $k$). Thus, our approach has inherently more tunable parameters. However, to ensure a fair comparison, we consider similar-sized tuning budgets for all methods. For this reason, we restrict our weight parameter $\alpha_m$ to either a constant value, $\alpha_m = 0.5$, or a decaying schedule defined by $\alpha_m = 0.9e^{-0.01m}$. The hyperparameter search spaces are presented in Table 2. Our approach was tuned using 72 configurations, XGBoost and its variants using 77 configurations, and TrAdaBoost.R2 using 72 configurations.

Table 2: Hyperparameter search spaces.

| Method | Base learner | Tree depth | Source tree depth | Learning rate | Scheduler | Estimators |
|---|---|---|---|---|---|---|
| TransferTreeBoost | Decision tree | $\{1, 2, 3\}$ | $\{1, 2, 3\}$ | $\{0.05, 0.1, 0.2, 0.3\}$ | $\alpha_m \in \{0.5, 0.9e^{-0.01m}\}$ | – |
| XGBoost | Decision tree | $\{1, \dots, 7\}$ | – | $\{0.05, 0.075, \dots, 0.3\}$ | – | – |
| XGBoost Warmstart | Decision tree | $\{1, \dots, 7\}$ | – | $\{0.05, 0.075, \dots, 0.3\}$ | – | – |
| Pooled XGBoost | Decision tree | $\{1, \dots, 7\}$ | – | $\{0.05, 0.075, \dots, 0.3\}$ | – | – |
| TrAdaBoost.R2 | M5Prime | $\{1, \dots, 4\}$ | – | $\{0.1, 0.5, 1.0\}$ | – | $\{10, 20, 30, 50, 100, 150\}$ |

### 5.3.2 Experimental setup

For all datasets, we employ a random 60/20/20 train/validation/test split on the target data, and we use early stopping after five consecutive non-improving iterations on the validation set. For XGBoost Warmstart, we also divide the source data into training and validation data using an 80/20 split, and apply an early stopping to obtain a base model, which is then used as the initial model for fine-tuning on the target data. For TrAdaBoost.R2, the implementation in the `Adapt` package does not support early stopping. Thus, we also tuned the number of estimators (iterations) for TrAdaBoost.R2 and omitted the use of the validation set. All experiments were repeated 30 times.

### 5.3.3 Results

The results for the best hyperparameter configurations are reported in Table 3. Each transfer technique was the best performer in three of the twelve scenarios. TransferTreeBoost performed on par with or better than XGBoost in all conducted experiments. In particular, it was the only transfer technique that showed this consistent behavior; XGBoost Warmstart, Pooled XGBoost, and TrAdaBoost.R2 degraded substantially under label shifts.

Table 3: Average RMSE values along with standard deviations in parenthesis.

| Dataset / Method | TransferTreeBoost | XGBoost | XGBoost Warmstart | Pooled XGBoost | TrAdaBoost.R2 |
|---|---|---|---|---|---|
| **UCI** | | | | | |
| InfraRed | 0.233 (0.034) | 0.235 (0.035) | 0.224 (0.033) | 0.216 (0.031) | **0.214** (0.034) |
| Concrete strength | 3.73 (1.001) | 3.885 (1.02) | **3.596** (0.512) | 3.866 (0.856) | 7.046 (1.028) |
| Auto MPG | 2.482 (0.502) | 2.489 (0.647) | 2.337 (0.499) | **2.245** (0.563) | 2.503 (0.699) |
| Real estate valuation | **4.873** (0.941) | 5.021 (0.918) | 5.221 (1.127) | 5.006 (0.923) | 5.408 (1.642) |
| Airfoil self-noise | 4.106 (0.259) | 4.139 (0.254) | **4.07** (0.281) | 4.108 (0.234) | 8.322 (0.406) |
| Forest fires | 88.174 (78.921) | 88.782 (78.737) | 85.602 (81.848) | 84.206 (82.74) | **83.701** (82.945) |
| **Stem volume** | | | | | |
| Pine → Spruce | 0.106 (0.022) | 0.119 (0.024) | 0.113 (0.03) | 0.105 (0.027) | **0.1** (0.021) |
| N. Sweden → S. Sweden | 0.12 (0.028) | 0.138 (0.033) | 0.138 (0.034) | **0.115** (0.03) | 0.124 (0.025) |
| Pine → Spruce with label shift | **0.103** (0.022) | 0.119 (0.024) | 1.695 (1.186) | 0.399 (5.781) | 75.067 (8.401) |
| **Remote sensing** | | | | | |
| Sweden → Latvia | 7.576 (1.744) | 7.667 (1.646) | **7.391** (1.526) | 7.513 (1.689) | 16.465 (2.002) |
| N. Sweden → Latvia | 7.605 (1.995) | 7.667 (1.646) | 7.623 (1.663) | **7.579** (1.688) | 16.419 (1.894) |
| Sweden → Latvia with label shift | **7.623** (2.222) | 7.667 (1.646) | 8.809 (7.176) | 12.946 (38.84) | 186.643 (20.701) |

For TransferTreeBoost, most transfer scenarios benefited from equal weighting of source/target tree contributions, *i.e.* $\alpha_m = 0.5$, compared to the decay scheduler defined by $\alpha_m = 0.9e^{-0.01m}$. Two scenarios based on splits of UCI datasets, Concrete strength and Airfoil self-noise, benefited from the decay scheduler, while the remaining scenarios benefited from equal weighting. The optimal hyperparameters of our approach are shown in Appendix H.2.

### 5.4 Friedman #1 revisited

We revisit the Friedman #1 dataset introduced in Section 4.1 to systematically evaluate the robustness of the considered methods under controlled domain shift. By creating source-target splits following the prescription in Section 4.1, we evaluate all approaches for five increasing levels of domain shift, controlled by $d \in \{3, 6, 9, 12, 15\}$. Each experiment uses 200 target instances and 1 000 source instances. We employ LSTransferTreeBoost without adding noise to the data, and reuse the hyperparameter search spaces from Section 5.3.1. The RMSE values were averaged over 10 seeds. The results for the best hyperparameter configurations are presented in Figure 3. TrAdaBoost.R2 performed poorly in this setting and is therefore omitted from the main figure; full results are provided in Appendix G.

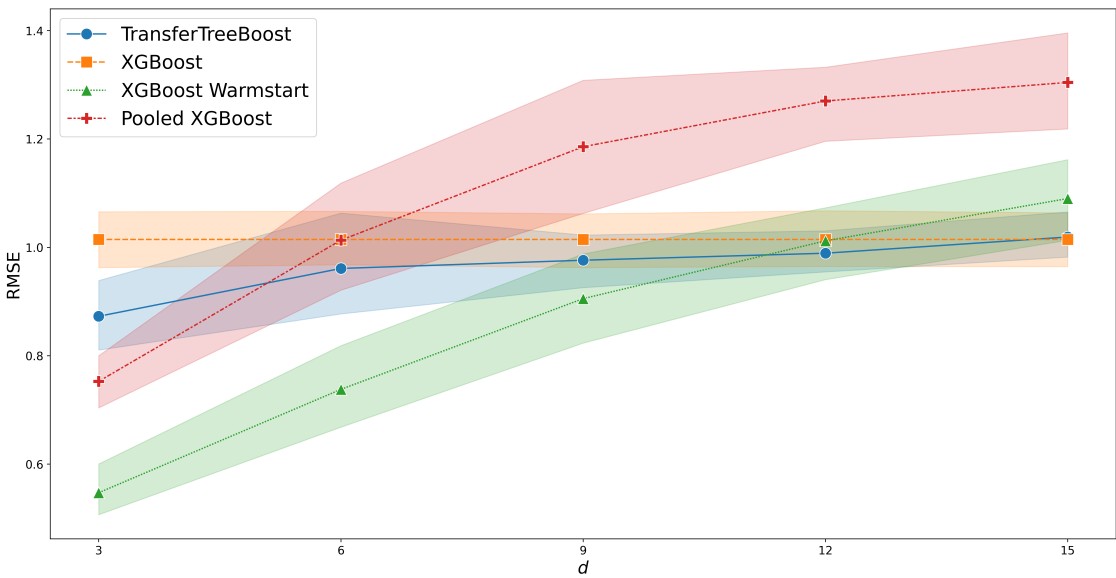

Figure 3: Performance on the Friedman #1 dataset under increasing domain shift ($d \in \{3, 6, 9, 12, 15\}$) using 1 000 source instances and 200 target instances, averaged over 10 seeds.

## 6 Discussion and future work

This work extends gradient tree boosting by directly integrating source-domain information into the boosting process, instead of depending on instance reweighting. This is achieved by jointly optimizing individual coefficients of a sum of a source tree and target tree on the target dataset. Consequently, source information is transferred implicitly by updating this combined base learner at each iteration.

We evaluated our approach against several established boosting-based regression transfer methods. Our findings indicate that while no single method achieves universal superiority, our approach performs on par with existing techniques and occasionally outperforms them. Notably, in all evaluated scenarios, it achieved results equal to or better than XGBoost, which is the only benchmark algorithm that, like our approach, solely performs optimization on target data. Importantly, while our approach makes use of source regions, model updates are driven exclusively by target data.

By updating only toward target data, our algorithm retains the convergence properties of gradient boosting and can be shown to minimize the training error at least as effectively as standard TreeBoost. Another important property obtained by the target-only optimization procedure is the potential of using relevant source data while ignoring irrelevant source information. This adaptive mechanism is important for mitigating negative transfer. In contrast, the other benchmark transfer algorithms often degraded substantially under larger domain shifts, such as severe label shifts. The robustness of our approach in these settings was further supported by a simulation study with increasing levels of domain shift (Section 5.4). As the domain shift increased, our method degraded more slowly relative to the compared baselines.

Instance-based methods can handle multiple source datasets by simply combining them into a single large dataset. Our current implementation supports only one source dataset, which (unless the source datasets are from similar distributions) limits its capabilities in multi-source settings. However, we show in Appendix C that optimal coefficients for our algorithm can also be derived in the multi-source setting. Further analyzing our algorithm in cases with multiple available source domains is an interesting direction for future work.

It also remains an open question whether systematic relationships between dataset characteristics and optimal hyperparameters exist beyond the settings considered here. A more comprehensive study of these relationships could potentially unravel such dependencies and support the development of functional default values. Another unexplored, yet relevant area of future research, is to assess performance relative to different numerical solvers and regularization techniques.

Our results suggest that gradient boosting provides an underexplored framework for transfer learning. In particular, model-based boosting transfer methods, such as TransferTreeBoost, retain theoretical guarantees while enabling direct knowledge transfer within the optimization process. We hope this work inspires future research along this avenue.

## Acknowledgments

This work was part of the ForestMap project, which is supported under the umbrella of ERA-NET Cofund ForestValue by the Swedish Governmental Agency for Innovation Systems, The Swedish Energy Agency, The Swedish Research Council for Environment, Agricultural Sciences and Spatial Planning, Academy of Finland, and The Scientific and Technological Research Council of Turkey. ForestValue has received funding from the European Union's Horizon 2020 research and innovation programme under grant agreement No. 773324. The research was also conducted within a collaborative Sweden–Brazil innovation initiative on AI-based forest inventory, funded by Vinnova, Sweden, under the international collaboration programme with Brazil in partnership with Federal University of Viçosa/EMBRAPII and Florestal Gateados.

Dag Björnberg's work was additionally funded by the Industry Graduate School on "Data Intensive Applications (DIA)" at Linnæus University, which is partially funded by The Knowledge Foundation, Sweden (project id 20190336).

Johan E.S. Fransson acknowledges the support from The Knowledge Foundation (project id 20200153), and The Bridge collaboration between Södra, IKEA, and Linnæus University.

The authors would like to thank Juris Zariņš and Latvian State Forest Research Institute (Silava) for providing Latvian reference data and Katam Technologies for providing the stem bank dataset. The authors would also like to thank Dr. Björn Lindenberg at Linnæus University and the three anonymous reviewers for their valuable comments and suggestions, which helped improve the manuscript.

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

# A  Proof of results from Section 3

*Proof of Theorem 1.* Throughout this proof we simplify notation by $h := h(\cdot, \mathbf{a}_m)$. Furthermore, denote by $\mathcal{H}_T$ and $\mathcal{H}_S$ the spaces of regression trees fitted on target and source data, respectively. In standard TreeBoost (Algorithm 1) the step $h$ is selected among the $J$-terminal node regression trees, *i.e.* $h \in \mathcal{H}_T$. In TransferTreeBoost, we select the base learner as a sum of trees, one from $\mathcal{H}_T$ and one from $\mathcal{H}_S$. This joint space is defined as

$$\mathcal{H}_{T+S} = \{h \colon h = w_1 h_t + w_2 h_s, h_t \in \mathcal{H}_T, h_s \in \mathcal{H}_S, w_1, w_2 \in \mathbb{R}\}.$$

Since the $K$-terminal tree

$$\sum_{i=1}^{K} 0 \cdot \mathbf{1}(\mathbf{x} \in \hat{R}_{km}),$$

exists in $\mathcal{H}_S$, we conclude that $\mathcal{H}_T \subset \mathcal{H}_{T+S}$. Hence, the problem can be viewed as a relaxation of the original TreeBoost optimization. Following the functional gradient descent interpretation of boosting (Mason et al., 1999), the convergence rate is governed by the alignment between the negative gradient and the best available direction in the hypothesis space. Since $\mathcal{H}_T \subset \mathcal{H}_{T+S}$, TransferTreeBoost is guaranteed to find a direction with alignment greater than or equal to that of standard TreeBoost, thereby ensuring a superior bound on the geometric convergence rate. This completes the proof of the theorem. $\qquad\square$

*Proof of Proposition 1.* Note that the sum of each row in $\mathbf{R}$ and $\mathbf{N}$ as well as $\hat{\mathbf{R}}$ and $\mathbf{N}^\mathsf{T}$ are equal, since each $\hat{R}_{im}$ are disjoint and $\cup_i \hat{R}_{im}$ is the entire ambient space, and the sum of the $i$th row in $\mathbf{N}$ is equal to the number of points in the region $R_{im}$, which trivially equals $|R_{im}|$. Hence, consider the vector $\mathbf{u} = [\mathbf{1}_J, -\mathbf{1}_K]^\mathsf{T}$, where the two components are vectors of length $J$ and $K$, respectively. Then

$$\mathbf{M}\mathbf{u} = [\mathbf{R}\mathbf{1}_J - \mathbf{N}\mathbf{1}_K, \mathbf{N}^\mathsf{T}\mathbf{1}_J - \hat{\mathbf{R}}\mathbf{1}_K]^\mathsf{T} = \mathbf{0}.$$

Since $\mathbf{u} \neq \mathbf{0}$ this implies that $\mathbf{M}$ is singular. This completes the proof of the proposition. $\qquad\square$

# B  Alternative view of a sum of two regression trees

We show that the sum of two regression trees can be expressed as a single equivalent regression tree. Consequently, the optimal coefficients could be derived using the same methods applied to single base learners, implying that the result is effectively just another regression tree.

**Lemma 1.** *Let $\mathbf{T}_1$ and $\mathbf{T}_2$ be two regression trees over the same feature space $X$ defined by regions and scalars $\{b_j, R_j\}_1^J$ and $\{\hat{b}_k, \hat{R}_k\}_1^K$, respectively. We define the sum of $\mathbf{T}_1$ and $\mathbf{T}_2$ as*

$$(\mathbf{T}_1 + \mathbf{T}_2)(\mathbf{x}) := \sum_{j=1}^{J} b_j \mathbf{1}(\mathbf{x} \in R_j) + \sum_{k=1}^{K} \hat{b}_k \mathbf{1}(\mathbf{x} \in \hat{R}_k). \tag{13}$$

*Then the corresponding sum is a regression tree.*

*Proof of Lemma 1.* Let $\mathbf{x}$ be an arbitrary datapoint. Applying the sum of the two trees to this datapoint gives us a prediction of the form of the RHS in (13). Since $\{R_j\}_{j=1}^J$ and $\{\hat{R}_k\}_{k=1}^K$ form two partitions of the same feature space, it follows that $\mathbf{x}$ must lie in the intersection of two of these sets ($\mathbf{x} \in \tilde{R}_{jk} := R_j \cap \hat{R}_k$ for some $j, k$). Thus, we can write

$$(\mathbf{T}_1 + \mathbf{T}_2)(\mathbf{x}) = \sum_{j,k} \tilde{b}_{jk} \mathbf{1}(\mathbf{x} \in \tilde{R}_{jk}), \tag{14}$$

where $\tilde{b}_{jk} = b_j + \hat{b}_k$. Since the regions $\tilde{R}_{jk}$ are disjoint and $\tilde{b}_{jk}$ is a scalar, it follows that the sum of the two trees is a new regression tree. $\qquad\square$

Figure 4 provides an illustration of the resulting equivalent single tree given from the sum of two regression trees.

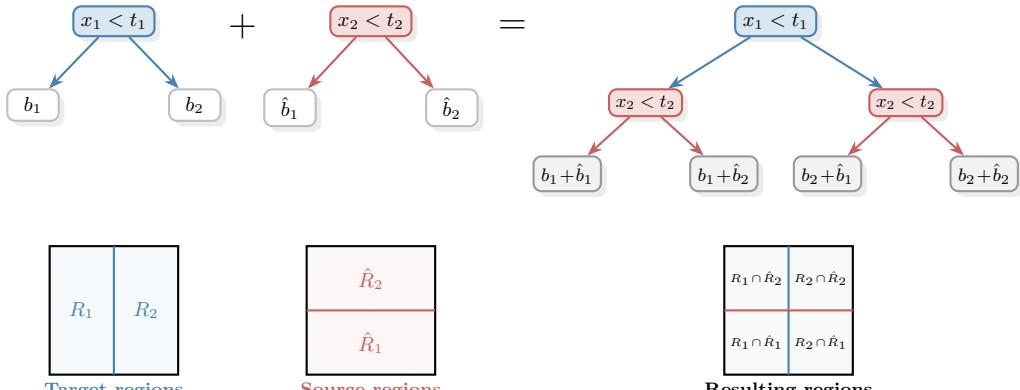

Figure 4: Visualization of Lemma 1. The partitions (below) illustrate how the regions overlap and make a finer grid.

## C   Multiple source domains

In a multi-source domain setting with $S$ source domains, the optimal coefficients at iteration $m$ are given by

$$\left\{\{\gamma_{jm}\}_{j=1}^{J}, \left\{\{\hat{\gamma}_{km}^{s}\}_{k=1}^{K(s)}\right\}_{s=1}^{S}\right\} =$$

$$\operatorname*{arg\,min}_{\left\{\{\gamma_{j}\}_{j=1}^{J}, \left\{\{\hat{\gamma}_{k}^{s}\}_{k=1}^{K(s)}\right\}_{s=1}^{S}\right\}} \sum_{i=1}^{N} L\left(y_i, F_{m-1}(\mathbf{x}_i) + \sum_{j=1}^{J}\gamma_j \mathbf{1}(\mathbf{x}_i \in R_{jm}) + \sum_{s=1}^{S}\sum_{k=1}^{K(s)}\hat{\gamma}_k^s \mathbf{1}(\mathbf{x}_i \in \hat{R}_{km}^s)\right). \qquad (15)$$

For notational convenience, let the target tree's regions and corresponding optimal coefficients be denoted by $\{\hat{R}_{km}^0\}_1^{K(0)}$ and $\{\hat{\gamma}_{km}^0\}_1^{K(0)}$, respectively, where $K(0) = J$. We can then write (15) as

$$\left\{\{\hat{\gamma}_{km}^{s}\}_{k=1}^{K(s)}\right\}_{s=0}^{S} = \operatorname*{arg\,min}_{\left\{\{\hat{\gamma}_{k}^{s}\}_{k=1}^{K(s)}\right\}_{s=0}^{S}} \sum_{i=1}^{N} L\left(y_i, F_{m-1}(\mathbf{x}_i) + \sum_{s=0}^{S}\sum_{k=1}^{K(s)}\hat{\gamma}_k^s \mathbf{1}(\mathbf{x}_i \in \hat{R}_{km}^s)\right). \qquad (16)$$

### C.1   LS regression

Denote by $B$ the objective in (16). Now, let $s' \in \{0, 1, 2, \dots S\}$ be an arbitrary dataset. We can write $B$ as the following sum

$$B = \sum_{k=1}^{K(s')} \sum_{x_i \in \hat{R}_{km}^{s'}} L\left(y_i, F_{m-1}(\mathbf{x}_i) + \hat{\gamma}_k^{s'} + \sum_{s\neq s'}\sum_{k=1}^{K(s)}\hat{\gamma}_k^s \mathbf{1}(\mathbf{x}_i \in \hat{R}_{km}^s)\right).$$

By differentiating $B$ w.r.t. $\hat{\gamma}_k^{s'}$ for $k = 1, 2, \dots, K(s')$, we get

$$\partial\hat{\gamma}_k^{s'} = \sum_{\mathbf{x}_i \in \hat{R}_{km}} \frac{\partial L}{\partial \hat{\gamma}_k^{s'}}.$$

If $L$ is the least-squares loss, the following condition must hold

$$\sum_{x_i \in \hat{R}_{km}^{s'}} \left(y_i - F_{m-1}(\mathbf{x}_i) - \hat{\gamma}_k^{s'} - \sum_{s\neq s'}\sum_{k=1}^{K(s)}\hat{\gamma}_k^s \mathbf{1}(\mathbf{x}_i \in \hat{R}_{km}^s)\right) = 0,$$

which is equivalent to

$$\sum_{x_i \in \hat{R}_{km}^{s'}} \left( \hat{\gamma}_k^{s'} + \sum_{s \neq s'} \sum_{k=1}^{K(s)} \hat{\gamma}_k^s \mathbf{1}(\mathbf{x}_i \in \hat{R}_{km}^s) \right) = \sum_{x_i \in \hat{R}_{km}^{s'}} (y_i - F_{m-1}(\mathbf{x}_i)). \tag{17}$$

Let $A$ and $B$ be two arbitrary regions. Denote by $|A|$ the number of datapoints in region $A$, and further denote by $N_{A \cap B}$ the number datapoints in the intersection of the two regions $A$ and $B$. Clearly,

$$\sum_{x_i \in \hat{R}_{km}^{s'}} \hat{\gamma}_k^{s'} = |\hat{R}_{km}^{s'}| \hat{\gamma}_k^{s'}.$$

Moreover, for a given region $\hat{R}_{k''m}^{s''}$ with $s'' \neq s'$, and corresponding coefficients $\hat{\gamma}_{k''}^{s''}$, we have that

$$\sum_{x_i \in \hat{R}_{km}^{s'}} \hat{\gamma}_{k''}^{s''} \mathbf{1}(\mathbf{x}_i \in \hat{R}_{k''m}^{s''}) = N_{\hat{R}_{km}^{s'} \cap \hat{R}_{k''m}^{s''}} \cdot \hat{\gamma}_{k''}^{s''}.$$

Thus, for each $k = 1, 2, \ldots, K(s')$, we can write (17) as

$$|\hat{R}_{km}^{s'}| \hat{\gamma}_k^{s'} + \sum_{s \neq s'} \sum_{k=1}^{K(s)} N_{\hat{R}_{km}^{s'} \cap \hat{R}_{km}^s} \cdot \hat{\gamma}_k^s = \sum_{x_i \in \hat{R}_{km}^{s'}} (y_i - F_{m-1}(\mathbf{x}_i)). \tag{18}$$

Now, define $\widehat{\mathbf{R}}^{s'} \in \mathbb{R}^{K(s') \times K(s')}$ and $\mathbf{r}^{s'} \in \mathbb{R}^{K(s')}$ as

$$\widehat{\mathbf{R}}^{s'} := \begin{bmatrix} |\hat{R}_{1m}^{s'}| & 0 & \cdots & 0 \\ 0 & |\hat{R}_{2m}^{s'}| & \cdots & 0 \\ \vdots & \vdots & \ddots & \vdots \\ 0 & 0 & \cdots & |\hat{R}_{K(s')m}^{s'}| \end{bmatrix}, \quad \text{and} \quad \mathbf{r}^{s'} := \begin{bmatrix} \sum_{\mathbf{x}_i \in \hat{R}_{1m}^{s'}} (y_i - F_{m-1}(\mathbf{x}_i)) \\ \sum_{\mathbf{x}_i \in \hat{R}_{2m}^{s'}} (y_i - F_{m-1}(\mathbf{x}_i)) \\ \vdots \\ \sum_{\mathbf{x}_i \in \hat{R}_{K(s')m}^{s'}} (y_i - F_{m-1}(\mathbf{x}_i)) \end{bmatrix}.$$

Let $\mathbf{N}^{s'} \in \mathbb{R}^{K(s') \times \sum_{s \neq s'} K(s)}$ be the matrix whose $k$th row has the elements of all intersections in (18). Note that $\mathbf{N}^{s'}$ contains pairwise intersections, and can thus be constructed from $S$ submatrices according to

$$\mathbf{N}^{s'} = \left[ \mathbf{N}_s^{s'} \right]_{s \in \{0, \ldots, S\} \setminus \{s'\}},$$

where $\mathbf{N}_s^{s'} \in \mathbb{R}^{K(s') \times K(s)}$, and can be written as

$$\mathbf{N}_s^{s'} = \begin{bmatrix} N_{\hat{R}_{1m}^{s'} \cap \hat{R}_{1m}^s} & N_{\hat{R}_{1m}^{s'} \cap \hat{R}_{2m}^s} & \cdots & N_{\hat{R}_{1m}^{s'} \cap \hat{R}_{K(s)m}^s} \\ N_{\hat{R}_{2m}^{s'} \cap \hat{R}_{1m}^s} & N_{\hat{R}_{2m}^{s'} \cap \hat{R}_{2m}^s} & \cdots & N_{\hat{R}_{2m}^{s'} \cap \hat{R}_{K(s)m}^s} \\ \vdots & \vdots & \ddots & \vdots \\ N_{\hat{R}_{K(s')m}^{s'} \cap \hat{R}_{1m}^s} & N_{\hat{R}_{K(s')m}^{s'} \cap \hat{R}_{2m}^s} & \cdots & N_{\hat{R}_{K(s')m}^{s'} \cap \hat{R}_{K(s)m}^s} \end{bmatrix}.$$

Combining all of this, we get the following linear system of equations

$$\begin{bmatrix} \widehat{\mathbf{R}}^0 & \mathbf{N}_1^0 & \mathbf{N}_2^0 & \cdots & \mathbf{N}_S^0 \\ \mathbf{N}_0^1 & \widehat{\mathbf{R}}^1 & \mathbf{N}_2^1 & \cdots & \mathbf{N}_S^1 \\ \mathbf{N}_0^2 & \mathbf{N}_1^2 & \widehat{\mathbf{R}}^2 & \cdots & \mathbf{N}_S^2 \\ \vdots & \vdots & \vdots & \ddots & \vdots \\ \mathbf{N}_0^S & \mathbf{N}_1^S & \mathbf{N}_2^S & \cdots & \widehat{\mathbf{R}}^S \end{bmatrix} \begin{bmatrix} \boldsymbol{\gamma}^0 \\ \boldsymbol{\gamma}^1 \\ \boldsymbol{\gamma}^2 \\ \vdots \\ \boldsymbol{\gamma}^S \end{bmatrix} = \begin{bmatrix} \mathbf{r}^0 \\ \mathbf{r}^1 \\ \mathbf{r}^2 \\ \vdots \\ \mathbf{r}^S \end{bmatrix},$$

or, by symmetry, the equivalent formulation

$$
\begin{bmatrix}
\widehat{\mathbf{R}}^0 & \mathbf{N}_1^0 & \mathbf{N}_2^0 & \cdots & \mathbf{N}_S^0 \\
\mathbf{N}_1^{0\mathsf{T}} & \widehat{\mathbf{R}}^1 & \mathbf{N}_2^1 & \cdots & \mathbf{N}_S^1 \\
\mathbf{N}_2^{0\mathsf{T}} & \mathbf{N}_2^{1\mathsf{T}} & \widehat{\mathbf{R}}^2 & \cdots & \mathbf{N}_S^2 \\
\vdots & \vdots & \vdots & \ddots & \vdots \\
\mathbf{N}_S^{0\mathsf{T}} & \mathbf{N}_S^{1\mathsf{T}} & \mathbf{N}_S^{2\mathsf{T}} & \cdots & \widehat{\mathbf{R}}^S
\end{bmatrix}
\begin{bmatrix}
\boldsymbol{\gamma}^0 \\ \boldsymbol{\gamma}^1 \\ \boldsymbol{\gamma}^2 \\ \vdots \\ \boldsymbol{\gamma}^S
\end{bmatrix}
=
\begin{bmatrix}
\mathbf{r}^0 \\ \mathbf{r}^1 \\ \mathbf{r}^2 \\ \vdots \\ \mathbf{r}^S
\end{bmatrix}.
$$

Hence, we may put

$$
\mathbf{M} := \begin{bmatrix}
\widehat{\mathbf{R}}^0 & \mathbf{N}_1^0 & \mathbf{N}_2^0 & \cdots & \mathbf{N}_S^0 \\
\mathbf{N}_1^{0\mathsf{T}} & \widehat{\mathbf{R}}^1 & \mathbf{N}_2^1 & \cdots & \mathbf{N}_S^1 \\
\mathbf{N}_2^{0\mathsf{T}} & \mathbf{N}_2^{1\mathsf{T}} & \widehat{\mathbf{R}}^2 & \cdots & \mathbf{N}_S^2 \\
\vdots & \vdots & \vdots & \ddots & \vdots \\
\mathbf{N}_S^{0\mathsf{T}} & \mathbf{N}_S^{1\mathsf{T}} & \mathbf{N}_S^{2\mathsf{T}} & \cdots & \widehat{\mathbf{R}}^S
\end{bmatrix},
$$

and analogous to Proposition 1, we have that $\mathbf{M}$ is singular as the vector

$$
\mathbf{w} := \frac{1}{S}\left[ S \cdot \mathbf{1}_{K(0)}, -\mathbf{1}_{K(1)}, -\mathbf{1}_{K(2)}, \ldots, -\mathbf{1}_{K(S)} \right]^{\mathsf{T}}
$$

serves as the vector $\mathbf{u}$ in the proof of Proposition 1, and the result of singularity follows by the same argument.

## C.2   LAD regression and Huber regression

Extending these approaches to multi-source settings is straight-forward and can be achieved in very similar ways. Consequently, we will only present the extension to multiple source domains for the least-absolute-deviation loss. Combining the reasoning for LAD regression with the arguments in Section 3.1.3, the extension to Huber regression in multi-source settings follows directly.

For the least-absolute-deviation loss, (16) becomes

$$
\left\{ \{\hat{\gamma}_{km}^s\}_{k=1}^{K(s)} \right\}_{s=0}^S = \underset{\left\{ \{\hat{\gamma}_k^s\}_{k=1}^{K(s)} \right\}_{s=0}^S}{\arg\min} \sum_{i=1}^N \left| y_i - F_{m-1}(\mathbf{x}_i) - \sum_{s=0}^S \sum_{k=1}^{K(s)} \hat{\gamma}_k^s \mathbf{1}(\mathbf{x}_i \in \hat{R}_{km}^s) \right|.
$$

Now, introduce auxiliary slack variables defined by

$$
t_i := \left| y_i - F_{m-1}(\mathbf{x}_i) - \sum_{s=0}^S \sum_{k=1}^{K(s)} \hat{\gamma}_k^s \mathbf{1}(\mathbf{x}_i \in \hat{R}_{km}^s) \right|, \quad i = 1, \ldots, N.
$$

Then we can formulate the search for the optimal coefficients as a linear programming (LP) problem as follows

$$
\min \sum t_i \tag{19}
$$
$$
\text{s.t.} \quad t_i \geq \left| y_i - F_{m-1}(\mathbf{x}_i) - \sum_{s=0}^S \sum_{k=1}^{K(s)} \hat{\gamma}_k^s \mathbf{1}(\mathbf{x}_i \in \hat{R}_{km}^s) \right|, \quad i = 1, \ldots, N.
$$

This is an optimization problem with $N + \sum_{s=0}^S K(s)$ unknown variables and $2N$ linear constraints. Let $\mathbf{I}_N$ be the identity matrix of size $N$, $\mathbf{c}_N$ be the identity vector of length $N$, $\mathbf{t}^{\mathsf{T}} = \left[ t_1 t_2 \ldots t_N \right]$ be the transpose

of the vector to the $N$ slack variables, and $(\mathbf{t}\hat{\boldsymbol{\gamma}})^\intercal = \left[\mathbf{t}^\intercal \hat{\gamma}_1^0 \hat{\gamma}_2^0 \ldots \hat{\gamma}_{K(0)}^0 \hat{\gamma}_1^1 \hat{\gamma}_2^1 \ldots \hat{\gamma}_{K(1)}^1 \ldots \hat{\gamma}_1^S \hat{\gamma}_2^S \ldots \hat{\gamma}_{K(S)}^S \right]$ be the transpose of the vector consisting of all decision variables (of length $N + \sum_{s=0}^{S} K(s)$). Furthermore, put

$$\boldsymbol{\Gamma} := \begin{bmatrix} \mathbf{1}(\mathbf{x}_1 \in \hat{R}_{1m}^0) & \cdots & \mathbf{1}(\mathbf{x}_1 \in \hat{R}_{K(0)m}^0) & \cdots & \mathbf{1}(\mathbf{x}_1 \in \hat{R}_{1m}^S) & \cdots & \mathbf{1}(\mathbf{x}_1 \in \hat{R}_{K(S)m}^S) \\ \mathbf{1}(\mathbf{x}_2 \in \hat{R}_{1m}^0) & \cdots & \mathbf{1}(\mathbf{x}_2 \in \hat{R}_{K(0)m}^0) & \cdots & \mathbf{1}(\mathbf{x}_2 \in \hat{R}_{1m}^S) & \cdots & \mathbf{1}(\mathbf{x}_2 \in \hat{R}_{K(S)m}^S) \\ \vdots & & \vdots & & \vdots & \ddots & \vdots \\ \mathbf{1}(\mathbf{x}_N \in \hat{R}_{1m}^0) & \cdots & \mathbf{1}(\mathbf{x}_N \in \hat{R}_{K(0)m}^0) & \cdots & \mathbf{1}(\mathbf{x}_N \in \hat{R}_{1m}^S) & \cdots & \mathbf{1}(\mathbf{x}_N \in \hat{R}_{K(S)m}^S) \end{bmatrix},$$

and

$$\mathbf{b} := \begin{bmatrix} y_1 - F_{m-1}(\mathbf{x}_1) \\ y_2 - F_{m-1}(\mathbf{x}_2) \\ \vdots \\ y_N - F_{m-1}(\mathbf{x}_N) \end{bmatrix}.$$

We can write (19) in matrix form as

$$\begin{aligned} \min \quad & \mathbf{c}^\intercal \mathbf{t} \\ \text{s.t.} \quad & \begin{bmatrix} -\mathbf{I}_N & \boldsymbol{\Gamma} \\ -\mathbf{I}_N & -\boldsymbol{\Gamma} \end{bmatrix} (\mathbf{t}\hat{\boldsymbol{\gamma}})^\intercal \leq \begin{bmatrix} \mathbf{b} \\ -\mathbf{b} \end{bmatrix}. \end{aligned} \tag{20}$$

Thus, we conclude that the optimal coefficients are given by solving the linear optimization in (20).

## D  Complexity of LAD and M

In this section, we seek to motivate the bounds of $\mathcal{C}_{\text{LAD}}$ and $\mathcal{C}_{\text{M}}$ in Section 3.1.4. We note that these may possibly be further improved, and we only analyze a single method to establish an upper bound. In particular, in this exposition we use an interior-point method, as outlined in Boyd and Vandenberghe (2004), to solve both the LP and QP problems.

Let $N$ denote the number of target data points, and denote by $J$ and $K$ the number of terminal regions in the target and source boosting trees. Further, we consider the LAD problem described in (8), and denote the matrices

$$\mathbf{A} := \begin{bmatrix} -\mathbf{I}_N & \boldsymbol{\Gamma} \\ -\mathbf{I}_N & -\boldsymbol{\Gamma} \end{bmatrix}, \quad \mathbf{b}_{\text{rhs}} := \begin{bmatrix} \mathbf{b} \\ -\mathbf{b} \end{bmatrix}, \quad \boldsymbol{\theta} := \begin{bmatrix} \boldsymbol{\gamma} \\ \hat{\boldsymbol{\gamma}} \end{bmatrix}, \quad \mathbf{u} := \begin{bmatrix} \mathbf{t} \\ \boldsymbol{\theta} \end{bmatrix}.$$

We further introduce a non-negative slack vector $\mathbf{s}$ to convert the inequalities into equality constraints, and a corresponding vector of non-negative Lagrange multipliers $\mathbf{y}$ for these constraints. Each of these vectors is of size $2N$.

We obtain the Lagrangian

$$\mathcal{L}(\mathbf{u}, \mathbf{s}, \mathbf{y}) = [\mathbf{c}^\intercal, \mathbf{0}^\intercal]^\intercal \mathbf{u} + \mathbf{y}^\intercal (\mathbf{A}\mathbf{u} + \mathbf{s} - \mathbf{b}_{\text{rhs}}).$$

The KKT stationarity condition with respect to $\mathbf{u}$ requires that the gradient of the Lagrangian vanishes, yielding

$$\nabla_{\mathbf{u}} \mathcal{L} = [\mathbf{c}^\intercal, \mathbf{0}^\intercal]^\intercal + \mathbf{A}^\intercal \mathbf{y} = \mathbf{0}.$$

Following Boyd and Vandenberghe (2004), we use a relaxed and perturbed formulation of the complementarity condition, where the right-hand side $\mu$ is allowed to be non-zero. We obtain the complementarity

$$\mathbf{s} \odot \mathbf{y} = \mu \mathbf{1},$$

where $\odot$ denotes the Hadamard product, and $\mu > 0$.

To track the central path toward the optimal solution, we form the Newton step by linearizing these KKT conditions. Let $\mathbf{S}$ and $\mathbf{Y}$ denote the diagonal matrices of $\mathbf{s}$ and $\mathbf{y}$, respectively. Further, put $\mathbf{r}_p := \mathbf{A}\mathbf{u} + \mathbf{s} - \mathbf{b}_{\text{rhs}}$ and $\mathbf{r}_d := \mathbf{A}^\intercal \mathbf{y} + [\mathbf{c}^\intercal, \mathbf{0}^\intercal]^\intercal$. We evaluate the system at the current iterate $(\mathbf{u}, \mathbf{y}, \mathbf{s})$ and seek a step

direction $(\Delta\mathbf{u}, \Delta\mathbf{y}, \Delta\mathbf{s})$ such that the updated point $(\mathbf{u} + \Delta\mathbf{u}, \mathbf{y} + \Delta\mathbf{y}, \mathbf{s} + \Delta\mathbf{s})$ satisfies the conditions. We obtain the following linearized system of equations

$$\begin{bmatrix} \mathbf{0} & \mathbf{A}^\intercal & \mathbf{0} \\ \mathbf{A} & \mathbf{0} & \mathbf{I} \\ \mathbf{0} & \mathbf{S} & \mathbf{Y} \end{bmatrix} \begin{bmatrix} \Delta\mathbf{u} \\ \Delta\mathbf{y} \\ \Delta\mathbf{s} \end{bmatrix} = \begin{bmatrix} -\mathbf{r}_d \\ -\mathbf{r}_p \\ -\mathbf{SY1} + \mu\mathbf{1} \end{bmatrix}. \tag{21}$$

Instead of solving the large system in (21) directly, we apply block elimination to exploit its structure. Solving for $\Delta\mathbf{s}$ in the second equation yields

$$\Delta\mathbf{s} = -\mathbf{A}\Delta\mathbf{u} - \mathbf{r}_p$$

and similarly from the third equation we obtain

$$\Delta\mathbf{y} = -\mathbf{Y1} + \mathbf{S}^{-1}\mu\mathbf{1} + \mathbf{S}^{-1}\mathbf{Y}\Delta\mathbf{s} = -\mathbf{Y1} + \mathbf{S}^{-1}\mu\mathbf{1} - \mathbf{S}^{-1}\mathbf{Y}(\mathbf{A}\Delta\mathbf{u} + \mathbf{r}_p).$$

Thus, $\Delta\mathbf{y}$ and $\Delta\mathbf{s}$ are described in terms of $\Delta\mathbf{u}$. By substituting for $\Delta\mathbf{y}$ in the first equation, we thus obtain the reduced system

$$\mathbf{A}^\intercal\Delta\mathbf{y} = -\mathbf{r}_d \iff \mathbf{A}^\intercal(-\mathbf{Y1} + \mathbf{S}^{-1}\mu\mathbf{1} - \mathbf{S}^{-1}\mathbf{Y}(\mathbf{A}\Delta\mathbf{u} + \mathbf{r}_p)) = -\mathbf{r}_d.$$

Simplifying the expression yields

$$-\mathbf{A}^\intercal\mathbf{S}^{-1}\mathbf{Y}\mathbf{A}\Delta\mathbf{u} = \mathbf{A}^\intercal(\mathbf{Y1} - \mathbf{S}^{-1}\mu\mathbf{1} + \mathbf{S}^{-1}\mathbf{Y}\mathbf{r}_p) - \mathbf{r}_d,$$

which may be further simplified to $(\mathbf{A}^\intercal\mathbf{S}^{-1}\mathbf{Y}\mathbf{A})\Delta\mathbf{u} = \tilde{\mathbf{r}}$, where $\tilde{\mathbf{r}}$ captures the negative of the right-hand side of the equation.

Put $\mathbf{W} := \mathbf{S}^{-1}\mathbf{Y}$, which we recall is a $2N \times 2N$ diagonal matrix. Because $\mathbf{s}$ and $\mathbf{y}$ represent the top and bottom halves of our $2N$ constraints, we partition $\mathbf{W}$ into two $N \times N$ diagonal blocks

$$\mathbf{W} = \begin{bmatrix} \mathbf{W}_1 & \mathbf{0} \\ \mathbf{0} & \mathbf{W}_2 \end{bmatrix}.$$

Thus, using our particular $\mathbf{A}$ we obtain

$$\mathbf{A}^\intercal\mathbf{W}\mathbf{A} = \begin{bmatrix} -\mathbf{I}_N & -\mathbf{I}_N \\ \mathbf{\Gamma}^\intercal & -\mathbf{\Gamma}^\intercal \end{bmatrix} \begin{bmatrix} \mathbf{W}_1 & \mathbf{0} \\ \mathbf{0} & \mathbf{W}_2 \end{bmatrix} \begin{bmatrix} -\mathbf{I}_N & \mathbf{\Gamma} \\ -\mathbf{I}_N & -\mathbf{\Gamma} \end{bmatrix}.$$

Multiplying these matrices out, yields

$$\begin{bmatrix} \mathbf{W}_1 + \mathbf{W}_2 & -(\mathbf{W}_1 - \mathbf{W}_2)\mathbf{\Gamma} \\ -\mathbf{\Gamma}^\intercal(\mathbf{W}_1 - \mathbf{W}_2) & \mathbf{\Gamma}^\intercal(\mathbf{W}_1 + \mathbf{W}_2)\mathbf{\Gamma} \end{bmatrix} \begin{bmatrix} \Delta\mathbf{t} \\ \Delta\boldsymbol{\theta} \end{bmatrix} = \begin{bmatrix} \tilde{\mathbf{r}}_{\mathbf{t}} \\ \tilde{\mathbf{r}}_{\boldsymbol{\theta}} \end{bmatrix}.$$

To solve this block system, we first eliminate $\Delta\mathbf{t}$ by solving the top block row. Because the top-left block $(\mathbf{W}_1 + \mathbf{W}_2)$ corresponds to the $N$ individual target variables, it is strictly diagonal and trivially inverted in $\mathcal{O}(N)$ operations. By substituting $\Delta\mathbf{t}$ into the bottom row, we obtain the Schur complement of the system with respect to $\Delta\boldsymbol{\theta}$. The reduced dense $(J + K) \times (J + K)$ system takes the form

$$\left(\mathbf{\Gamma}^\intercal(\mathbf{W}_1 + \mathbf{W}_2)\mathbf{\Gamma} - \mathbf{\Gamma}^\intercal(\mathbf{W}_1 - \mathbf{W}_2)(\mathbf{W}_1 + \mathbf{W}_2)^{-1}(\mathbf{W}_1 - \mathbf{W}_2)\mathbf{\Gamma}\right)\Delta\boldsymbol{\theta} = \tilde{\mathbf{r}}'.$$

Since all inner weight matrices are diagonal, they commute. We may thus factor out the $\mathbf{\Gamma}$ matrices to express the reduced system as $(\mathbf{\Gamma}^\intercal\widetilde{\mathbf{W}}\mathbf{\Gamma})\Delta\boldsymbol{\theta} = \tilde{\mathbf{r}}'$. Here, $\widetilde{\mathbf{W}}$ is a single $N \times N$ collapsed diagonal weight matrix defined as

$$\widetilde{\mathbf{W}} := (\mathbf{W}_1 + \mathbf{W}_2) - (\mathbf{W}_1 - \mathbf{W}_2)^2(\mathbf{W}_1 + \mathbf{W}_2)^{-1}.$$

Because $\widetilde{\mathbf{W}}$ is diagonal, evaluating the product $\mathbf{\Gamma}^\intercal\widetilde{\mathbf{W}}\mathbf{\Gamma}$ algebraically reduces into computing a weighted sum of $N$ outer products, each formed by a $(J + K)$-dimensional row of $\mathbf{\Gamma}$. This matrix construction incurs a dominant computational cost of $\mathcal{O}(N(J+K)^2)$. Finally, solving this dense symmetric positive-definite system

via Cholesky factorization requires $\mathcal{O}((J + K)^3)$ operations. Summing these costs and multiplying by the number of iterations $I$ yields the stated complexity bound of $\mathcal{C}_{\mathrm{LAD}}(N, J, K) \in \mathcal{O}(I(N(J+K)^2 + (J+K)^3))$.

Furthermore, the Huber regression formulated in Section 3.1.3 is a QP problem, and the proof follows the same principle as the LP formulation, mutatis mutandis. However, we provide the details for the interested reader. Let $\mathbf{Q}_N$ and $\mathbf{c}_N$ denote the diagonal weight matrix and linear cost vector for the slack variables $\mathbf{t}$, as defined in (10). In an interior-point framework, the Newton step for this QP modifies the KKT stationarity condition, replacing the linear objective gradient with the augmented vector $[\mathbf{c}_N^\mathsf{T}, \mathbf{0}^\mathsf{T}]^\mathsf{T} + \mathbf{Qu}$. Here, the Hessian is strictly block-diagonal with $\mathbf{Q} = \mathrm{diag}(\mathbf{Q}_N, \mathbf{0})$, as there is no quadratic penalty on the parameters $\boldsymbol{\theta}$. Consequently, the unreduced KKT matrix is modified solely by the addition of $\mathbf{Q}_N$ to the top-left block. This top-left block becomes $(\mathbf{Q}_N + \mathbf{W}_1 + \mathbf{W}_2)$. Because $\mathbf{Q}_N$ is strictly diagonal, this sum remains strictly diagonal and is trivially invertible in $\mathcal{O}(N)$ operations. The Schur complement formation and the subsequent Cholesky factorization proceed exactly as in the LP case, yielding the matching worst-case complexity of $\mathcal{C}_{\mathrm{M}}(N, J, K) \in \mathcal{O}(I(N(J+K)^2 + (J+K)^3))$.

# E   Wall-clock runtimes

We provide empirical runtimes on the Friedman #1 dataset and its altered version with $d = 5$ (see Section 4.1) for all three regression approaches using the current implementation. The source dataset consisted of 1 000 instances, while the number of target instances were in the set $\{100, 200, \ldots, 1\,000\}$. The source tree depth and the target tree depth were both set to 2. We then performed 100 boosting iterations and averaged runtime across 50 experiments. The runtimes are shown in Figure 5. Note that default solver configurations were used for solving the LP and QP problems, and these may be unaware of the specific block structure of the KKT matrix, which is why the particular reductions made in Appendix D do not necessarily apply. This motivates the empirical nonlinear growth in $N$ for the LAD and M variants of the coefficient updates.

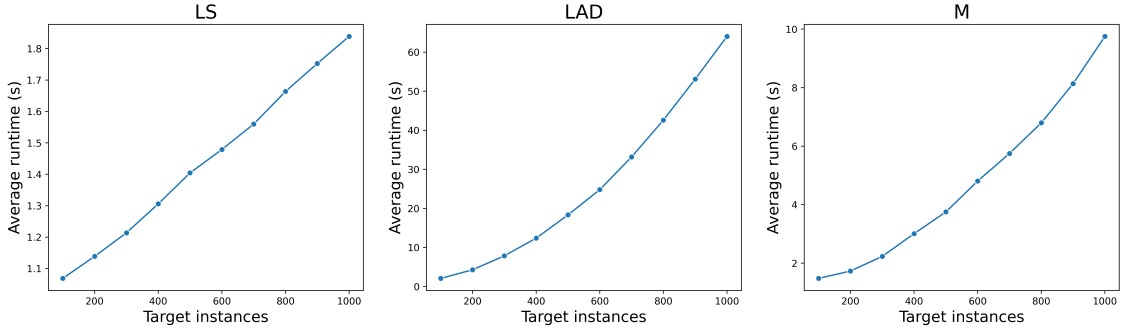

Figure 5: Average wall-clock runtimes for LSTransferTreeBoost (LS), LADTransferTreeBoost (LAD), and MTransferTreeBoost (M) on the Friedman #1 dataset and its altered version ($d = 5$) across increasing numbers of target instances. Experiments were conducted using a source dataset of 1 000 instances, with both source and target tree depths fixed at 2. Each method is run for 100 boosting iterations, and runtimes are averaged over 50 independent runs.

# F   Benchmark comparison with DL

To compare the boosting algorithms detailed in Sections 3 and 5.2 against a relevant state-of-the-art deep learning method for tabular data, we include a comparison with two ResNet configurations: a target-only model, and a model pretrained on the source data before being finetuned on the target data. We adopted the tabular ResNet implementation proposed by Gorishniy et al. (2021), as it was robust across a range of benchmarks.

Both ResNet variants were tuned over an identical hyperparameter space, with learning rates $\in$ $\{1e{-}3, 1e{-}4, 5e{-}4\}$, dropout rates $\in \{0, 0.1, 0.2\}$, hidden sizes $\in \{64, 128, 256\}$, and numbers of blocks $\in \{2, 3, 4\}$, for a total of 81 hyperparameter configurations. The target-only model was trained following the procedure in Section 5.3.2, using a 60/20/20 train/validation/test split and early stopping based on validation performance. The finetuned model was first pretrained on source data using an 80/20 train/validation split, with early stopping applied after five consecutive non-improving iterations.

The results are presented in Table 4. For reference, we include RMSE values for XGBoost, as well as for the best boosting-based model in each transfer scenario.

Table 4: Average RMSE values along with standard deviations in parenthesis.

| Dataset / Method | ResNet (target-only) | ResNet (finetuned) | XGBoost | Best Boosting Model |
|---|---|---|---|---|
| InfraRed | 0.222 (0.027) | 0.216 (0.024) | 0.235 (0.035) | **0.214** (0.034) |
| Concrete strength | 6.821 (1.228) | 5.652 (1.103) | 3.885 (1.02) | **3.596** (0.512) |
| Auto MPG | 2.55 (0.852) | 2.421 (0.674) | 2.489 (0.647) | **2.245** (0.563) |
| Real estate valuation | 5.343 (1.573) | 5.055 (1.331) | 5.021 (0.918) | **4.873** (0.941) |
| Airfoil self-noise | 4.109 (0.33) | 4.103 (0.349) | 4.139 (0.254) | **4.07** (0.281) |
| Forest fires | 68.836 (68.665) | **63.017** (71.72) | 88.782 (78.737) | 83.701 (82.945) |
| **Stem volume** | | | | |
| Pine $\to$ Spruce | 0.102 (0.024) | **0.096** (0.024) | 0.119 (0.024) | 0.1 (0.021) |
| N. Sweden $\to$ S. Sweden | 0.119 (0.037) | **0.104** (0.029) | 0.138 (0.033) | 0.115 (0.029) |
| Pine $\to$ Spruce with label shift | 0.101 (0.025) | **0.097** (0.023) | 0.119 (0.024) | 0.103 (0.022) |
| **Remote sensing** | | | | |
| Sweden $\to$ Latvia | 8.473 (3.376) | 7.487 (1.752) | 7.667 (1.491) | **7.391** (1.466) |
| N. Sweden $\to$ Latvia | 8.152 (2.244) | **7.347** (2.379) | 7.667 (1.491) | 7.579 (1.554) |
| Sweden $\to$ Latvia with label shift | 8.298 (2.461) | **7.564** (1.614) | 7.667 (1.491) | 7.623 (1.567) |

## G  Results on the Friedman #1 dataset

We present a complete figure of the experiments performed in Section 5.4 (with TrAdaBoost.R2 included). The results are shown in Figure 6.

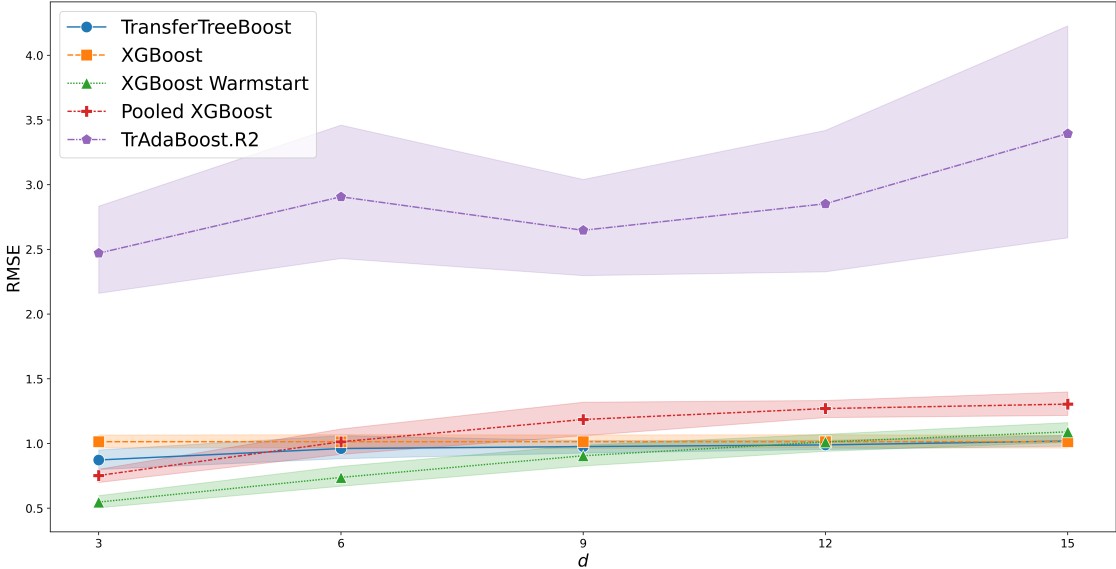

Figure 6: Performance on the Friedman #1 dataset under increasing domain shift ($d \in \{3, 6, 9, 12, 15\}$) using 1 000 source instances and 200 target instances, averaged over 10 seeds.

## H   Optimal hyperparameters

### H.1   Friedman #1

We present the top-5 hyperparameter configurations for MTransferTreeBoost for each triplet (target instances, error distribution, $d$), for the experiments conducted in Section 4.2. These are shown in Tables 5, 6, and 7.

Table 5: Top-5 hyperparameter configurations for each target instance and $d$ for Gaussian errors.

| Target instances | $d$ | Max. source tree depth | Max. target tree depth | $m_0$ | $k$ | avg. RMSE |
|---|---|---|---|---|---|---|
| 100 | 3 | 1 | 1 | 0.9 | 0.01 | 1.937 |
| | | 1 | 1 | 0.5 | 0.0 | 1.969 |
| | | 2 | 1 | 0.1 | 0.0 | 1.98 |
| | | 2 | 1 | 0.5 | 0.01 | 1.982 |
| | | 3 | 1 | 0.1 | 0.0 | 1.986 |
| | 6 | 3 | 1 | 0.9 | 0.05 | 2.03 |
| | | 2 | 1 | 0.9 | 0.05 | 2.04 |
| | | 3 | 1 | 0.1 | 0.05 | 2.047 |
| | | 3 | 1 | 0.1 | 0.0 | 2.049 |
| | | 3 | 1 | 0.5 | 0.05 | 2.051 |
| | 9 | 3 | 1 | 0.5 | 0.05 | 2.068 |
| | | 3 | 1 | 0.5 | 0.01 | 2.082 |
| | | 3 | 2 | 0.5 | 0.0 | 2.09 |
| | | 3 | 2 | 0.5 | 0.01 | 2.092 |
| | | 3 | 1 | 0.9 | 0.05 | 2.092 |
| 300 | 3 | 2 | 2 | 0.9 | 0.01 | 1.279 |
| | | 3 | 1 | 0.1 | 0.0 | 1.299 |
| | | 3 | 1 | 0.5 | 0.01 | 1.307 |
| | | 3 | 1 | 0.9 | 0.05 | 1.314 |
| | | 3 | 2 | 0.9 | 0.01 | 1.316 |
| | 6 | 3 | 1 | 0.1 | 0.0 | 1.316 |
| | | 2 | 1 | 0.5 | 0.0 | 1.35 |
| | | 1 | 1 | 0.5 | 0.0 | 1.351 |
| | | 2 | 2 | 0.9 | 0.01 | 1.351 |
| | | 3 | 1 | 0.5 | 0.0 | 1.354 |
| | 9 | 1 | 1 | 0.5 | 0.0 | 1.352 |
| | | 3 | 1 | 0.9 | 0.01 | 1.361 |
| | | 3 | 1 | 0.9 | 0.05 | 1.364 |
| | | 3 | 1 | 0.5 | 0.0 | 1.364 |
| | | 2 | 1 | 0.5 | 0.0 | 1.37 |
| 500 | 3 | 3 | 2 | 0.9 | 0.05 | 1.118 |
| | | 2 | 2 | 0.9 | 0.01 | 1.131 |
| | | 2 | 2 | 0.5 | 0.0 | 1.135 |
| | | 3 | 2 | 0.5 | 0.01 | 1.143 |
| | | 2 | 2 | 0.5 | 0.01 | 1.144 |
| | 6 | 2 | 2 | 0.5 | 0.0 | 1.134 |
| | | 3 | 2 | 0.5 | 0.0 | 1.143 |
| | | 3 | 2 | 0.5 | 0.01 | 1.144 |
| | | 3 | 2 | 0.9 | 0.05 | 1.149 |
| | | 2 | 2 | 0.9 | 0.01 | 1.154 |
| | 9 | 2 | 2 | 0.9 | 0.01 | 1.152 |
| | | 1 | 2 | 0.5 | 0.0 | 1.153 |
| | | 3 | 2 | 0.9 | 0.01 | 1.154 |
| | | 2 | 2 | 0.5 | 0.0 | 1.164 |
| | | 1 | 2 | 0.9 | 0.01 | 1.175 |

Table 6: Top-5 hyperparameter configurations for each target instance and $d$ for slash errors.

| Target instances | $d$ | Max. source tree depth | Max. target tree depth | $m_0$ | $k$ | avg. RMSE |
|---|---|---|---|---|---|---|
| 100 | 3 | 2 | 2 | 0.9 | 0.01 | 1.557 |
| | | 2 | 1 | 0.9 | 0.01 | 1.562 |
| | | 3 | 2 | 0.9 | 0.01 | 1.575 |
| | | 2 | 1 | 0.5 | 0.01 | 1.583 |
| | | 3 | 1 | 0.9 | 0.01 | 1.593 |
| | 6 | 2 | 1 | 0.5 | 0.01 | 1.614 |
| | | 3 | 1 | 0.9 | 0.01 | 1.62 |
| | | 3 | 1 | 0.5 | 0.01 | 1.622 |
| | | 3 | 1 | 0.5 | 0.0 | 1.629 |
| | | 2 | 1 | 0.9 | 0.01 | 1.629 |
| | 9 | 2 | 1 | 0.5 | 0.01 | 1.687 |
| | | 1 | 1 | 0.5 | 0.0 | 1.689 |
| | | 2 | 1 | 0.5 | 0.0 | 1.691 |
| | | 1 | 1 | 0.9 | 0.01 | 1.692 |
| | | 2 | 1 | 0.9 | 0.01 | 1.697 |
| 300 | 3 | 3 | 2 | 0.9 | 0.01 | 0.985 |
| | | 3 | 3 | 0.9 | 0.01 | 0.985 |
| | | 1 | 2 | 0.9 | 0.01 | 0.996 |
| | | 1 | 2 | 0.9 | 0.05 | 1.013 |
| | | 3 | 2 | 0.9 | 0.05 | 1.013 |
| | 6 | 2 | 2 | 0.5 | 0.0 | 1.029 |
| | | 1 | 2 | 0.5 | 0.01 | 1.038 |
| | | 2 | 2 | 0.9 | 0.01 | 1.04 |
| | | 1 | 2 | 0.9 | 0.01 | 1.04 |
| | | 2 | 2 | 0.5 | 0.01 | 1.041 |
| | 9 | 1 | 2 | 0.9 | 0.01 | 1.053 |
| | | 1 | 2 | 0.5 | 0.0 | 1.054 |
| | | 1 | 2 | 0.5 | 0.01 | 1.056 |
| | | 2 | 2 | 0.9 | 0.01 | 1.062 |
| | | 1 | 2 | 0.9 | 0.05 | 1.062 |
| 500 | 3 | 2 | 2 | 0.9 | 0.01 | 0.779 |
| | | 1 | 2 | 0.9 | 0.01 | 0.783 |
| | | 3 | 2 | 0.9 | 0.01 | 0.791 |
| | | 3 | 2 | 0.5 | 0.01 | 0.792 |
| | | 3 | 3 | 0.9 | 0.01 | 0.794 |
| | 6 | 2 | 2 | 0.9 | 0.01 | 0.784 |
| | | 3 | 2 | 0.5 | 0.0 | 0.786 |
| | | 3 | 2 | 0.5 | 0.01 | 0.793 |
| | | 2 | 2 | 0.5 | 0.0 | 0.796 |
| | | 3 | 2 | 0.9 | 0.05 | 0.802 |
| | 9 | 3 | 2 | 0.9 | 0.01 | 0.795 |
| | | 3 | 2 | 0.9 | 0.05 | 0.8 |
| | | 2 | 2 | 0.9 | 0.01 | 0.802 |
| | | 1 | 2 | 0.9 | 0.01 | 0.817 |
| | | 3 | 2 | 0.5 | 0.0 | 0.82 |

Table 7: Top-5 hyperparameter configurations for each target instance and $d$ for $t(2)$-errors.

| Target instances | $d$ | Max. source tree depth | Max. target tree depth | $m_0$ | $k$ | avg. RMSE |
|---|---|---|---|---|---|---|
| 100 | 3 | 3 | 1 | 0.9 | 0.01 | 1.537 |
| | | 3 | 1 | 0.5 | 0.01 | 1.546 |
| | | 3 | 1 | 0.5 | 0.0 | 1.55 |
| | | 2 | 1 | 0.9 | 0.01 | 1.568 |
| | | 2 | 1 | 0.5 | 0.01 | 1.571 |
| | 6 | 3 | 1 | 0.9 | 0.05 | 1.601 |
| | | 3 | 1 | 0.5 | 0.01 | 1.629 |
| | | 3 | 1 | 0.5 | 0.05 | 1.63 |
| | | 2 | 1 | 0.9 | 0.05 | 1.643 |
| | | 3 | 1 | 0.1 | 0.0 | 1.647 |
| | 9 | 3 | 1 | 0.1 | 0.01 | 1.658 |
| | | 2 | 1 | 0.5 | 0.0 | 1.662 |
| | | 2 | 1 | 0.9 | 0.01 | 1.664 |
| | | 2 | 1 | 0.5 | 0.01 | 1.676 |
| | | 3 | 1 | 0.9 | 0.05 | 1.678 |
| 300 | 3 | 3 | 2 | 0.9 | 0.01 | 1.074 |
| | | 3 | 3 | 0.9 | 0.01 | 1.089 |
| | | 3 | 1 | 0.5 | 0.0 | 1.113 |
| | | 1 | 2 | 0.9 | 0.01 | 1.115 |
| | | 3 | 2 | 0.5 | 0.0 | 1.119 |
| | 6 | 3 | 2 | 0.9 | 0.01 | 1.127 |
| | | 3 | 2 | 0.5 | 0.0 | 1.147 |
| | | 3 | 2 | 0.9 | 0.05 | 1.152 |
| | | 2 | 2 | 0.9 | 0.01 | 1.156 |
| | | 3 | 3 | 0.9 | 0.01 | 1.158 |
| | 9 | 3 | 2 | 0.9 | 0.05 | 1.164 |
| | | 1 | 2 | 0.9 | 0.01 | 1.171 |
| | | 3 | 2 | 0.1 | 0.0 | 1.179 |
| | | 2 | 2 | 0.1 | 0.0 | 1.187 |
| | | 3 | 2 | 0.5 | 0.0 | 1.188 |
| 500 | 3 | 3 | 2 | 0.5 | 0.0 | 0.888 |
| | | 3 | 2 | 0.9 | 0.01 | 0.891 |
| | | 3 | 2 | 0.5 | 0.01 | 0.895 |
| | | 3 | 2 | 0.9 | 0.05 | 0.9 |
| | | 3 | 3 | 0.9 | 0.01 | 0.902 |
| | 6 | 3 | 2 | 0.9 | 0.01 | 0.88 |
| | | 3 | 2 | 0.5 | 0.01 | 0.888 |
| | | 3 | 2 | 0.5 | 0.0 | 0.892 |
| | | 2 | 2 | 0.9 | 0.01 | 0.897 |
| | | 3 | 2 | 0.9 | 0.05 | 0.9 |
| | 9 | 3 | 2 | 0.9 | 0.01 | 0.891 |
| | | 3 | 2 | 0.5 | 0.0 | 0.899 |
| | | 1 | 2 | 0.5 | 0.0 | 0.918 |
| | | 2 | 2 | 0.5 | 0.0 | 0.92 |
| | | 3 | 2 | 0.5 | 0.01 | 0.922 |

## H.2 Real datasets

We present the optimal hyperparameters for each dataset in Section 5.3.

Table 8: Optimal hyperparameter configurations for LSTransferTreeBoost for each transfer scenario.

| Dataset / Hyperparameter | $\nu$ | Max. target tree depth | Max. source tree depth | $m_0$ | $k$ |
|---|---|---|---|---|---|
| **UCI** | | | | | |
| InfraRed | 0.05 | 2.0 | 2.0 | 0.5 | 0.0 |
| Concrete strength | 0.3 | 3.0 | 3.0 | 0.9 | 0.01 |
| Auto MPG | 0.2 | 1.0 | 3.0 | 0.5 | 0.0 |
| Real estate valuation | 0.2 | 2.0 | 1.0 | 0.5 | 0.0 |
| Airfoil self-noise | 0.2 | 2.0 | 2.0 | 0.9 | 0.01 |
| Forest fires | 0.05 | 1.0 | 2.0 | 0.5 | 0.0 |
| **Stem volume** | | | | | |
| Pine $\rightarrow$ Spruce | 0.2 | 2.0 | 1.0 | 0.5 | 0.0 |
| N. Sweden $\rightarrow$ S. Sweden | 0.05 | 2.0 | 2.0 | 0.5 | 0.0 |
| Pine $\rightarrow$ Spruce with label shift | 0.2 | 2.0 | 3.0 | 0.5 | 0.0 |
| **Remote sensing** | | | | | |
| Sweden $\rightarrow$ Latvia | 0.2 | 2.0 | 2.0 | 0.5 | 0.0 |
| N. Sweden $\rightarrow$ Latvia | 0.3 | 2.0 | 2.0 | 0.5 | 0.0 |
| Sweden $\rightarrow$ Latvia with label shift | 0.3 | 2.0 | 1.0 | 0.5 | 0.0 |

