# OpenReview forum: "Gradient Tree Boosting for Regression Transfer"
_TMLR — Accepted by TMLR_

### Review · Reviewer_QwES · 2026-03-03

**Summary Of Contributions:**

This paper proposes a model-based transfer boosting method for regression, where each boosting iteration uses a sum of two trees, one trained on the target domain and one on the source domain. The authors further derive how to estimate the region-wise coefficients under different losses, such as LS / LAD / Huber, and report results on simulated UCI-style transfer settings and two real forestry datasets.

**Audience:**

Yes

**Audience Explanation:**

The idea of injecting source-domain tree structures into boosting is interesting, and if the transfer mechanism can be clarified and compared fairly with warm-start and existing transfer boosting methods, the findings could be useful for practitioners who work with small target datasets.

**Claims And Evidence:**

No

**Claims Explanation:**

The paper provides empirical results showing consistent gains over a target-only XGBoost baseline on several tasks, and the method is described with algorithmic steps and derivations. However, the current evidence is not fully convincing for transfer learning claims, mainly due to limited baselines, unclear mechanism explanation, and missing complexity and stability analysis.

**Requested Changes:**

- It's recommended to revise the related work and citations. Some references like Huber loss and robust regression are not presented in a standard way, and the authors should use more canonical and easily verifiable sources.

- The authors should clarify the mechanism. For example, in Algorithm 2 (Step 5) the source gradients are computed, while Eq.(3) optimizes coefficients by summing only over the target samples. It is not clear why a tree fitted on source gradients can reliably help on target data, and what assumptions are needed for this to work.

- For the claimed “model-based boosting transfer” contribution, the authors should carefully explain these questions clearly:
     - In Appendix B (Lemma 1), the sum of two trees is mathematically equivalent to a single tree defined on intersected regions. This suggests the method is not a new boosting paradigm, but more like fitting a more complex partition while reusing source split structures as a structural prior/regularization. The authors should discuss this explicitly and explain where the real novelty lies beyond re-parameterization.
     - Warm-start or continuation training for boosting models is widely used in practice, like one trains a source model and then continues boosting on the target data. The authors should clearly explain how their method differs from warm-start in principle, and provide a direct experimental comparison.
    - The paper cites “Adapted tree boosting for transfer learning”. The authors should provide a more detailed discussion of the differences in mechanisms, and it is better to include experimental comparisons to support the claimed advantages.

- The authors mention using NumPy solvers or adding a small ridge penalty to handle matrix singularity. The authors should discuss solver dependence and stability, for example, do different solvers or different regularization strengths lead to different solutions and different performance? It is recommended to add a sensitivity study to improve reproducibility.

- For LAD/Huber versions, each boosting iteration solves an LP/QP with N slack variables. The authors should provide time and memory complexity discussion, and report training time versus performance improvements, so readers can understand the trade-off.

- The experimental section needs strengthening:
    - Even if TrAdaBoost.R2 performs worse, it is better to still include classic transfer baselines to make the evaluation complete. Otherwise, it is hard to tell whether the gains come from transfer learning, or simply from adding structural regularization/increased model flexibility.

    - The hyperparameter tuning is not balanced, the proposed method is tuned over several parameters, while XGBoost seems not tuned with a comparable budget. The authors should use a similar random search budget for XGBoost to ensure a fair comparison and robustness.

    - Similar to point 5, the computational cost looks high. The authors should report runtime and memory, otherwise it is difficult to judge practicality.

    - The paper lacks key ablations, e.g., results with a_m = 0 and other simple settings, to show the contribution of each component.

    - The UCI transfer setting is artificially constructed by splitting. The authors should add experiments where the domain shift is more realistic (or at least discuss limitations), because the current setup may not reflect real distribution shifts.

---

> ### Author Response · Authors · 2026-04-11
>
> Dear reviewer,
>
> Thank you for your valuable review. We are especially thankful for your comments regarding providing a mechanism explanation and adding more experiments with realistic domain shifts. These proposals helped us make substantial improvements to our research. We have tried to address these comments (along with the other points) as carefully as possible in the revised version.
>
> Sincerely,
>
> The authors
>
> Requested changes:
> * _It's recommended to revise the related work and citations. Some references like Huber loss and robust regression are not presented in a standard way, and the authors should use more canonical and easily verifiable sources._
>
> Response: We have made changes to Section 3.1 and 3.1.3., e.g., we provide a reference to Huber’s original work. The naming of the Huber regression as M-regression was to be consistent with Friedman’s original work on gradient boosting machines. We have replaced the term M-regression with Huber regression, but our method is still called MTransferTreeBoost under the Huber loss, to remain consistent with Friedman. We now state this explicitly. We hope these changes are in line with the reviewer’s request.
>
> * _The authors should clarify the mechanism. For example, in Algorithm 2 (Step 5) the source gradients are computed, while Eq.(3) optimizes coefficients by summing only over the target samples. It is not clear why a tree fitted on source gradients can reliably help on target data, and what assumptions are needed for this to work._
>
> Response: We thank the reviewer for pointing out the need for this justification. The assumption of our approach is that the source and target domains share informative partition boundaries in the feature space. Mechanically, the tree fitted on the source information acts as a structural prior. Together with the target tree it proposes a candidate partition of the feature space. In Equation (3), we obtain a projection of the target data onto this proposed partition. Because the optimization of $\hat{\gamma}$ is performed strictly on target samples, the algorithm possesses an adaptive mechanism, which can upweight or downweight the coefficients depending on the degree of domain mismatch. The first part of this response is now included in the manuscript in Section 3, and a simulation showing the adaptive mechanism is added in Section 4.4.
>
> * _For the claimed “model-based boosting transfer” contribution, the authors should carefully explain these questions clearly:_
>     * _In Appendix B (Lemma 1), the sum of two trees is mathematically equivalent to a single tree defined on intersected regions. This suggests the method is not a new boosting paradigm, but more like fitting a more complex partition while reusing source split structures as a structural prior/regularization. The authors should discuss this explicitly and explain where the real novelty lies beyond re-parameterization._
>
>     Response: We agree with the characterization of our method: we also view the resulting source tree regions acting precisely as a structural prior (Section 3). The idea of using the sum of two trees as our base learner is to be able to control the weight of the source tree, which appears beneficial in some cases compared to equal weighting, and to make it clear how much that source information contributes to the overall optimization. It is not clear to us how this can be done on the equivalent single tree defined on intersected regions. Our contribution in terms of boosting is to use a sum of trees (with derived coefficient updates), in regression transfer settings.
>
>     * _Warm-start or continuation training for boosting models is widely used in practice, like one trains a source model and then continues boosting on the target data. The authors should clearly explain how their method differs from warm-start in principle, and provide a direct experimental comparison._
>
>     Response: Indeed, warmstarting XGBoost is widely used in practice, and we have added it as a benchmark (along with pooled XGBoost). See Section 5.2.
>
>     * _The paper cites “Adapted tree boosting for transfer learning”. The authors should provide a more detailed discussion of the differences in mechanisms, and it is better to include experimental comparisons to support the claimed advantages._
>
>     Response: The referred paper was included merely as an example of a transfer version of XGBoost and deals with classification tasks. As we understand it, the method described in the referenced paper is initialized by training a model on the source dataset (as in warmstart boosting). Secondly, the model is transferred to the target domain, where certain operations are performed on the base model, such as re-weighting of leaves, pruning rare branches, and performing new splits to accommodate the target dataset. We believe that this text is not necessary to include in the introduction, but we keep it as a reference.

---

> > ### Author Response · Authors · 2026-04-11
> >
> > * _The authors mention using NumPy solvers or adding a small ridge penalty to handle matrix singularity. The authors should discuss solver dependence and stability, for example, do different solvers or different regularization strengths lead to different solutions and different performance? It is recommended to add a sensitivity study to improve reproducibility._
> >
> > Response: Analyzing solver dependency or regularization techniques is, in our opinion, beyond the scope of this paper. Although we agree that this is yet to be fully explored, this is left for future work (see Section 6). However, we do agree there is a need for reproducibility, which is why our full repo including necessary details and the used packages with versions is available alongside the paper.
> >
> > * _For LAD/Huber versions, each boosting iteration solves an LP/QP with N slack variables. The authors should provide time and memory complexity discussion, and report training time versus performance improvements, so readers can understand the trade-off._
> >
> > Response: This requested change was addressed as a general comment to all reviewers.
> >
> > * _The experimental section needs strengthening:_
> >     * _Even if TrAdaBoost.R2 performs worse, it is better to still include classic transfer baselines to make the evaluation complete. Otherwise, it is hard to tell whether the gains come from transfer learning, or simply from adding structural regularization/increased model flexibility._
> >
> >     Response: This requested change was addressed as a general comment to all reviewers.
> >
> >     * _The hyperparameter tuning is not balanced, the proposed method is tuned over several parameters, while XGBoost seems not tuned with a comparable budget. The authors should use a similar random search budget for XGBoost to ensure a fair comparison and robustness._
> >
> >     Response: This requested change was addressed as a general comment to all reviewers.
> >
> >     * _Similar to point 5, the computational cost looks high. The authors should report runtime and memory, otherwise it is difficult to judge practicality._
> >
> >     Response: This requested change was addressed as a general comment to all reviewers.
> >
> >     * _The paper lacks key ablations, e.g., results with a_m = 0 and other simple settings, to show the contribution of each component._
> >
> >     Response: Because our work concerns modeling the base learner as a sum of two trees, it makes sense to include both trees in the model update.
> >
> >     * _The UCI transfer setting is artificially constructed by splitting. The authors should add experiments where the domain shift is more realistic (or at least discuss limitations), because the current setup may not reflect real distribution shifts._
> >
> >     Response: Indeed, the artificial splits of UCI data may not reflect real transfer scenarios. However, splitting strategies are commonly used for evaluation purposes. We explicitly state this limitation in Section 5.1.1. Finding real, challenging benchmark datasets for tabular transfer learning proved non-trivial. We decided to use the two real datasets we had at our disposal and add two more realistic transfer scenarios for each dataset (see Section 5.1.2 and 5.1.3). Thus, the empirical validation is now performed on 8 datasets in 12 transfer scenarios.

---

### Review · Reviewer_Gffg · 2026-03-10

**Summary Of Contributions:**

This paper designs an extension of gradient tree boosting for regression in a transfer learning setting where the target dataset is small but related source data is available. While existing transfer learning methods reweight source samples, this work designs a model-based approach named  TransferTreeBoost, where the boosting base learner is the sum of two regression trees, with one trained on target data and one on source data. Their combinations are jointly optimized for the target task.

This work derives update rules for several loss functions (least-squares, least-absolute-deviation, and Huber loss) and designs a weighting parameter that controls how much the source tree influences each boosting iteration.

Simulation experiments are conducted on UCI datasets and real forestry applications. The results show that the proposed algorithm consistently improves prediction accuracy compared with a strong baseline (XGBoost trained only on target data).

**Audience:**

Yes

**Audience Explanation:**

The work addresses transfer learning for regression on tabular datasets. This problem is common in many real-world machine learning applications where labeled target data is scarce but related source datasets are available.

This work extends gradient tree boosting with a principled way to incorporate source-domain information directly into the boosting process rather than relying on instance reweighting. This work is relevant to transfer learning, boosting methods, and tabular machine learning.

**Claims And Evidence:**

Yes

**Claims Explanation:**

- The paper provides a mathematical formulation of the proposed TransferTreeBoost algorithm and also derives the optimization procedures for several loss functions (least-squares, least-absolute-deviation, and Huber). This work provides theoretical results that the algorithm achieves a convergence rate at least as good as the classical boosting tree algorithm.

- The authors evaluate the method using synthetic datasets where they systematically vary factors such as target dataset size, error distributions, and degree of domain shift between source and target domains. This controlled setup tests how the algorithm behaves under different settings. The results show that the Huber-based variant performs well under heavy-tailed noise.

- The method is also evaluated on multiple datasets, including UCI datasets and two forestry applications. The results show that TransferTreeBoost achieves lower RMSE than the baseline model (XGBoost trained only on target data) in seven out of eight datasets.

**Requested Changes:**

- It would be better to include more baseline methods for comparison. The experiments mainly compare the proposed method with XGBoost trained on target data only. There are additional transfer learning baselines, such as two-stage TrAdaBoost.R2 (ICML 2010), or other transfer methods for tabular data.

Pardoe, David, and Peter Stone. Boosting for regression transfer. International Conference on Machine Learning 2010.

- It would be better to provide more discussion of computational complexity and runtime. The time complexity of the algorithm is not analyzed for the LAD and Huber variants, where solving LP or QP problems is required.

- It would be better to discuss the extension to multiple source domains. The current algorithm focuses on a single source dataset, while many transfer learning scenarios involve multiple sources. Further including empirical evaluations on multiple source domains would be interesting.

---

> ### Author Response · Authors · 2026-04-11
>
> Dear reviewer,
>
> Thank you for your feedback and for your positive assessment of our work. We have carefully considered your comments regarding benchmarking, computational complexity, and multiple source domains in the revision.
>
> Sincerely,
>
> The authors
>
> Requested changes:
> * _It would be better to include more baseline methods for comparison. The experiments mainly compare the proposed method with XGBoost trained on target data only. There are additional transfer learning baselines, such as two-stage TrAdaBoost.R2 (ICML 2010), or other transfer methods for tabular data._
>
> Response: This requested change was addressed as a general comment to all reviewers.
>
> * _It would be better to provide more discussion of computational complexity and runtime. The time complexity of the algorithm is not analyzed for the LAD and Huber variants, where solving LP or QP problems is required._
>
> Response: This requested change was addressed as a general comment to all reviewers.
>
> * _It would be better to discuss the extension to multiple source domains. The current algorithm focuses on a single source dataset, while many transfer learning scenarios involve multiple sources. Further including empirical evaluations on multiple source domains would be interesting._
>
> Response: We agree that empirical results for multiple source domains would be an interesting addition. However, our primary focus in this paper is to introduce the new method and evaluate it in the most straightforward way possible. For this reason, we have limited the extension for multiple domains ($S\geq2$) in the Appendix to a purely theoretical description. We state in Section 6 that this is an interesting direction for future work.

---

### Review · Reviewer_MoTL · 2026-03-29

**Summary Of Contributions:**

**Summary**

The paper proposes a transfer-learning extension of gradient tree boosting. At each boosting step, the weak learner is modeled as the sum of two trees, one fit on source data and one fit on target data, with both optimized for the target task. The authors derive the coefficients update under three loss functions and report that, in their benchmarks, the method outperforms XGBoost in most cases.

**Strengths**
 - The paper is clearly written and well organized.
 - The method is developed for three loss functions, which broadens its applicability across different regression settings and noise regimes.

**Weaknesses**
 - Foundational concern: tree-based methods are not generally regarded as strong transfer-learning models--the paper itself cites literature suggesting this. As a result, the motivation remains incomplete: it is not clear whether the proposed method is mainly of methodological interest or whether it is also practically competitive. Either way, without comparison to modern transfer-learning approaches, it is difficult to assess the significance and practical relevance of the proposed method.
 - Empirical validation is very limited. There is no comparison against a transfer-learning method for tabular data--the only baseline is target-only XGBoost, and even that comparison is hard to interpret because the proposed method is tuned over a richer hyperparameter space than XGBoost. There is no analysis of failure modes.
 - Computational analysis is missing. If the method is meant as an alternative to XGBoost, runtime and scalability matter, especially since the LAD and Huber variants involve LP/QP-style optimization.

**Additional Comments:**

Regarding the schedule for $\alpha_m$: what is the intuition for using a decaying schedule rather than, for example, an increasing one? Why should the model rely more heavily on the source tree in early boosting rounds? Is there empirical evidence that decay performs better than the reverse schedule or other alternatives?

**Audience:**

No

**Audience Explanation:**

Although the technical construction may be of niche interest, the paper does not clearly establish whether its main contribution is conceptual or practical, and it provides no evidence of how the method compares with existing transfer-learning approaches for tabular data.

**Broader Impact Concerns:**

None.

**Claims And Evidence:**

No

**Claims Explanation:**

The main claim appears to be that gradient boosting provides a "powerful [...] framework for transfer learning on tabular data". I believe the experiments do not substantiate this claim--see requested changes below. With the current evidence, a more defensible claim would be: gradient-boosted trees, which are not typically strong transfer-learning models, can be modified to become more transfer-aware. However, it remains unclear how competitive the resulting method is relative to SOTA transfer-learning methods for tabular data, and therefore how useful it is in practice.

**Requested Changes:**

- **1.** *[required]* Clarify the paper’s intent. Is the contribution primarily a methodological/theoretical exercise showing how to adapt boosting to a setting it is not naturally suited for, or is it meant to propose a practically competitive transfer-learning method? Either framing is acceptable, but both require comparisons with modern transfer learning methods.

 - **2.** *[required]* Add deep-learning baselines. I separate this from the broader request below because I think is essential to establish where the method stands relative to current best practice. It is perfectly fine if the proposed method does not compete with SOTA, but without such comparisons the paper’s practical significance and intended contribution remain unclear. The authors should at least include: target-only deep models (e.g. FT-Transformer or ResNet), the corresponding source-pretrained/target-finetuned model, and a transfer-oriented tabular architecture.

 - **3.** *[required]* Further expand the experiments by: (a) including additional baselines, such as fine-tuned XGBoost and pooled (source + target) XGBoost; (b) making the XGBoost comparison fairer by tuning XGBoost over a modest but standard hyperparameter space beyond depth alone, for example learning rate, number of boosting rounds (with early stopping), and one or two regularization or subsampling parameters; and (c) explicitly testing failure modes, such as negative transfer/extreme source-target mismatch. For (c), the authors could systematically vary source-target mismatch, include one clearly misaligned source domain, and report when TransferTreeBoost falls below target-only performance. Since the method introduces a source-weighting parameter, it would also be interesting to analyze whether different \alpha_m schedules mitigate this degradation.

 - **4.** *[required]* Discuss computational complexity and runtime.

---

> ### Author Response · Authors · 2026-04-11
>
> Dear reviewer,
>
> Thank you for your valuable input. We appreciated your comments regarding the intent of the paper, which helped us clarify our contribution and improve the manuscript. Moreover, your suggestions on studying varying source-target domain mismatch were especially helpful for improving our work. We have carefully considered all of your comments and addressed them to the best of our ability.
>
> Sincerely,
>
> The authors
>
> Requested changes:
> * _1. [required] Clarify the paper’s intent. Is the contribution primarily a methodological/theoretical exercise showing how to adapt boosting to a setting it is not naturally suited for, or is it meant to propose a practically competitive transfer-learning method? Either framing is acceptable, but both require comparisons with modern transfer learning methods._
>
> Response: The intent of the paper was never to establish a new SOTA approach for tabular data regression transfer, but rather to extend gradient tree boosting for transfer learning. We agree that some formulations in the manuscript could be interpreted as if we are proposing a new SOTA approach. Therefore, we have removed some of the text to avoid giving the reader this impression.
>
> * _2. [required] Add deep-learning baselines. I separate this from the broader request below because I think is essential to establish where the method stands relative to current best practice. It is perfectly fine if the proposed method does not compete with SOTA, but without such comparisons the paper’s practical significance and intended contribution remain unclear. The authors should at least include: target-only deep models (e.g. FT-Transformer or ResNet), the corresponding source-pretrained/target-finetuned model, and a transfer-oriented tabular architecture._
>
> Response: While we acknowledge the reviewer’s suggestion to include deep-learning baselines, we note that the focus of this work is on transfer learning within the gradient boosting framework. In this setting, approaches such as XGBoost Warmstart and Pooled XGBoost are widely used in practice (see Section 5.2.2)  and constitute the most relevant and standard comparison class for our method. Given this scope, we do not consider deep-learning baselines essential for evaluating the contributions of this paper. That said, we recognize that such models could provide additional context for completeness, and we are willing to include them in a revised version if the reviewer and AE consider it necessary.
>
> * _3. [required] Further expand the experiments by: (a) including additional baselines, such as fine-tuned XGBoost and pooled (source + target) XGBoost; (b) making the XGBoost comparison fairer by tuning XGBoost over a modest but standard hyperparameter space beyond depth alone, for example learning rate, number of boosting rounds (with early stopping), and one or two regularization or subsampling parameters; and (c) explicitly testing failure modes, such as negative transfer/extreme source-target mismatch. For (c), the authors could systematically vary source-target mismatch, include one clearly misaligned source domain, and report when TransferTreeBoost falls below target-only performance. Since the method introduces a source-weighting parameter, it would also be interesting to analyze whether different \alpha_m schedules mitigate this degradation._
>
> Response: a) We have included additional baselines, including XGBoost Warmstart, Pooled XGBoost, and TrAdaBoost.R2. b) We now use comparable budget sizes for all evaluated approaches. Regarding XGBoost (and its variants) we have added a search space for the shrinkage parameter. We do not consider other hyperparameters, such as subsampling or regularization parameters. These can, in principle, also be added to our method, and adding such parameters could make it more difficult to compare the methods. c) We thank the reviewer for this suggestion. We argue that our method is robust under large domain shifts, see Remark 1. We also demonstrated this robustness in Section 5.4, where we systematically varied source-target mismatch. Further, we have also added two realistic transfer scenarios (Section 5.1.2 and 5.1.3) with severe label shift. The results implied that our method was robust under this type of shift relative to the compared baselines.
>
> * _4. [required] Discuss computational complexity and runtime._
>
> Response: We agree that this is a flaw of the original version of the paper. We have added a section to discuss the complexity and derive complexity bounds. Further, we include some simple runtime examples from our implementation for the various losses to give the reader an indication of what to expect. However, we do not provide a full run-time comparison as e.g., XGBoost is highly optimized, whereas our current implementation has not undergone any major optimization efforts, which would make such a direct comparison unbalanced.

---

> > ### Author Response · Authors · 2026-04-11
> >
> > Additional comment:
> > * _Regarding the schedule for $\alpha_m$ what is the intuition for using a decaying schedule rather than, for example, an increasing one? Why should the model rely more heavily on the source tree in early boosting rounds? Is there empirical evidence that decay performs better than the reverse schedule or other alternatives?_
> >
> > Response: In the results of the simulation study (Section 4.3), we added some text on that static $\alpha_m$ values of 0.9 (highly weighted source tree contribution) resulted in poor performance. The reverse schedule (increasing) was tested in early stages of this work, and did not perform well. Thus, we never included it in the paper.

---

> > > ### Comment · Reviewer_MoTL · 2026-04-17
> > >
> > > Thank you for the revisions. The paper has benefited from the additional experiments and baselines, which provide a more reasonable empirical evaluation. At the same time, with the expanded baselines, the results (and thus the claims) are more credible but also noticeably weaker. Also, many of the improvements reported in Table 3 appear small relative to the observed variance, and are therefore unlikely to be statistically significant.
> > >
> > > Regarding deep-learning baselines: I appreciate the clarification of the paper’s intent and the effort to better scope the contribution. However, I do not find the argument for excluding such baselines convincing. Even if the goal is not to establish a new state-of-the-art method, the paper still proposes a transfer learning approach. Its practical relevance cannot be assessed in isolation from current standard approaches in that setting. For this reason, I consider the inclusion of representative deep-learning baselines necessary to properly contextualize the method.
> > >
> > > Given these concerns, I am not yet convinced of the paper’s practical significance.

---

> > > > ### Author Response · Authors · 2026-04-23
> > > >
> > > > Dear reviewer,
> > > >
> > > > Thank you for your comments. We still consider boosting-based methods to be the most relevant comparison class, as our approach is developed within the same methodological framework. Our primary goal is therefore to assess performance within this family of methods. That said, we understand that comparisons to deep-learning approaches can provide useful additional context. In response to your suggestion, we have included results for a target-only ResNet baseline, as well as a transfer learning variant (train on source, refine on target). While these models are not the primary focus of our study, we agree that their inclusion helps contextualize our method in a better way. To maintain the focus of the main text, these results are reported in Appendix F, with a reference provided in Section 5.2.

---

### Author Response · Authors · 2026-04-11

Dear AE and reviewers,

We want to express our gratitude to the AE for his efforts and to the three reviewers for providing high-quality feedback. Indeed, the reviewers provided valuable input, which helped us make substantial improvements to the manuscript.

We identified three points shared by all three reviewers, that we have addressed accordingly.


* More benchmark algorithms: We now include other benchmark boosting-based algorithms. Our rationale for using target-only XGBoost as the sole baseline model was that it is a strong non-transfer baseline built on gradient tree boosting. However, we acknowledge the need for more benchmarks. Therefore, we have included additional boosting-based approaches, including XGBoost Warmstart, Pooled XGBoost, and TrAdaBoost.R2. For the TrAdaBoost.R2 algorithm, we discovered a bug in our previous implementation, causing the bad predictions. We have now resolved this, and indeed, this is a relevant baseline.

* Time complexity analysis: We provide a theoretical time complexity analysis of our algorithm under all loss functions. To be precise, we provide upper bounds for the coefficient update operations. We do not report a comparison on wall-clock runtime, because that would result in unfair and inaccurate comparisons. For instance, the XGBoost package is highly optimized, while our current  implementation is not.  However, in the appendix, we report run-time for our three different losses on the Friedman dataset.

* Similar-sized tuning budgets for all compared baselines: Since our approach has more tunable parameters compared to XGBoost, it is not certain that similar-sized budgets will result in a fairer comparison. However, we acknowledge the reviewers’ concern about potentially over-optimistic RMSE values for our approach due to the larger budget. To address this, we now tune each approach using comparable budgets.

For more detailed and directed responses, we have addressed each reviewer separately.

Sincerely,

The authors

---

> ### Author Response · Authors · 2026-04-17
> **Updated code available**
>
> Dear AE and reviewers,
>
> The code including the additional experiments in the revision is now available.
>
> Sincerely,
>
> The authors

---

> ### Author Response · Authors · 2026-04-23
> **New revision available**
>
> Dear AE and reviewers,
>
> We have submitted a new revision to address a comment by reviewer MoTL concerning deep learning baselines (changes are in Section 5.2 and Appendix F). The code has also been updated.
>
> Sincerely,
>
> The authors

---

### Decision · Action_Editor_ohQ8 · 2026-05-14

**Recommendation:** Accept as is

**Audience:**

Yes

**Audience Explanation:**

The paper is likely to be of interest to researchers working on tabular transfer learning, especially those interested in boosting-based methods and regression transfer. The proposed framework and technical formulation provide relevant methodological insights for the community, even if the empirical gains are limited.

**Claims And Evidence:**

Yes

**Claims Explanation:**

The paper provides a technically sound methodological extension of gradient tree boosting for regression transfer and presents a clear algorithmic framework that supports multiple loss functions. The empirical evaluation appears consistent with the claims made, although the experimental results do not demonstrate a significant improvement over competing methods. In addition, questions remain regarding scalability and the practical advantages of the approach in real-world settings. Overall, the evidence is adequate to support the methodological claims, but less convincing regarding practical superiority.